# Structures of liganded glycosylphosphatidylinositol transamidase illuminate GPI-AP biogenesis

Yidan Xu [1,3], Tingting Li [1,3], Zixuan Zhou[2,3], Jingjing Hong[1], Yulin Chao[2], Zhini Zhu[2], Ying Zhang[2], Qianhui Qu [2] ✉ & Dianfan Li [1] ✉

Many eukaryotic receptors and enzymes rely on glycosylphosphatidylinositol (GPI) anchors for membrane localization and function. The transmembrane complex GPI-T recognizes diverse proproteins at a signal peptide region that lacks consensus sequence and replaces it with GPI via a transamidation reaction. How GPI-T maintains broad specificity while preventing unintentional cleavage is unclear. Here, substrates- and products-bound human GPI-T structures identify subsite features that enable broad proprotein specificity, inform catalytic mechanism, and reveal a multilevel safeguard mechanism against its promiscuity. In the absence of proproteins, the catalytic site is invaded by a locally stabilized loop. Activation requires energetically unfavorable rearrangements that transform the autoinhibitory loop into crucial catalytic cleft elements. Enzyme-proprotein binding in the transmembrane and luminal domains respectively powers the conformational rearrangement and induces a competent cleft. GPI-T thus integrates various weak specificity regions to form strong selectivity and prevent accidental activation. These findings provide important mechanistic insights into GPI-anchored protein biogenesis.

Glycosylphosphatidylinositol (GPI) anchoring is a post-translational modification highly conserved in all eukaryots[1–4]. Human cells encode over 150 GPI-anchored proteins (GPI-APs) including cell surface receptors, complement regulators, transcytotic transporters, enzymes/inhibitors, and adhesion molecules. These proteins play critical roles in various biological processes such as embryogenesis, neurodevelopment, tumorigenesis, and immunomodulation (reviewed in ref. 1–4). Insufficient GPI-AP synthesis due to genetic defects can lead to severe developmental diseases, and the upregulation of GPI-AP biogenesis enzymes has been reported in cancers[1,3,4]. Some GPI-APs also serve as biomarkers for diseases such as cancer[5] (e.g. carcinoembryonic antigen), hepatic injury[6] (e.g. alkaline phosphatase), and male fertility[7] (e.g. TEX101). Moreover, pathogens such as *Trypanosoma brucei*, responsible for the fetal sleeping sickness, deploy GPI-APs to evade host adaptive immunity[8]. The biosynthesis pathway of GPI-APs in these pathogens is thus a validated target for antiparasitic[9] and antimycotic[10] drugs.

GPI-AP biogenesis represents a metabolic expensive pathway that involves over 20 intramembrane catalytic steps (Supplementary Fig. 1a)[2,11]. Phosphatidylinositol is first modified by glucosamine (GlcN), mannoses (Man), and ethanolamine phosphates (EtNP) to produce GPI, typically characterized by a complex glycan core of α-Man3-(1 → 2)-α-Man2-(1 → 6)-α-Man1-(1 → 4)-α-GlcN (Supplementary Fig. 1b)[12]. The glycolipid is then added to the proproteins by the GPI

[1]State Key Laboratory of Molecular Biology, CAS Center for Excellence in Molecular Cell Science, Shanghai Institute of Biochemistry and Cell Biology, Chinese Academy of Sciences (CAS), University of CAS, Shanghai, China. [2]Shanghai Stomatological Hospital, School of Stomatology, Shanghai Key Laboratory of Medical Epigenetics, International Co-laboratory of Medical Epigenetics and Metabolism (Ministry of Science and Technology), Institutes of Biomedical Sciences, Department of Systems Biology for Medicine, Fudan University, Shanghai, China. [3]These authors contributed equally: Yidan Xu, Tingting Li, Zixuan Zhou. ✉e-mail: qqh@fudan.edu.cn; dianfan.li@sibcb.ac.cn

transamidase (GPI-T), a transmembrane complex composed of five subunits: GAAP1 (Gaa1p), PIGK (Gpi8p), PIGS (Gpi17p), PIGT (Gpi16p), and PIGU (Gab1p) (yeast homologs in brackets)[1–4]. The so formed GPI-APs undergoes a subsequent series of remodeling steps before being transported to the cell surface[1,11].

The GPI-T complex is a promiscuous enzyme. First, GPI-T exhibits broad proprotein specificity. For example, the human GPI-T complex can process over 150 proproteins ranging from <20 to >2,000 residues. GPI-T recognizes proproteins through a remarkably vague pattern in the C-terminal signal peptide (CSP) region rather than consensus sequences. The pattern consists of an ω-site where GPI is later added, followed by an ω+1 site that can be any residue except proline, a small ω+2 residue, a generally hydrophilic spacer with 8-12 residues, and a stretch of 15-20 hydrophobic residues for membrane association (Fig. 1a). An unstructured linker of approximately 10 polar residues (ω−10 to ω−1) proceeding the CSP is also found in GPI-APs[1–4]. The ω-site typically contains residues with small side chains that include glycine, alanine, serine, asparagine, and aspartate[1–4], but is recently[13] expanded to include two slightly larger residues (leucine,

methionine) and at a Cβ-branched amino acid (threonine), albeit at a lower frequency (Fig. 1a). Second, GPI-T exhibits promiscuity for the GPI substrate. For example, EtNP3 on GPI is long thought to be the sole physiological bridge for GPI attachment, but a recent study[14] shows EtNP2 is the preferred choice for some GPI-APs. What is more, GPI-T exhibits activities with non-GPI amines including hydroxylamine[15]. Finally, GPI-T can digest proproteins without GPI attachment[16]. Given GPI-T's broad proprotein specificity, its promiscuity raises an important question as how its activity is safeguarded to prevent unintentional cleavage.

Genetic defects of GPI-T subunits cause severe NEDHCAS (neurodevelopmental disorders with hypotonia and cerebellar atrophy, with or without seizures)[17–21], while abnormal amplification of GPI-T subunits are linked to cancers[22,23]. Recent structural studies have provided insights into the mechanisms underlying these defects[24,25], apart from revealing the overall assembly of the GPI-T complex and the GPI-binding cavity. However, due to the absence of a proprotein in these structures, several crucial mechanistic questions remain regarding the structural basis for proprotein recognition and broad specificity, the

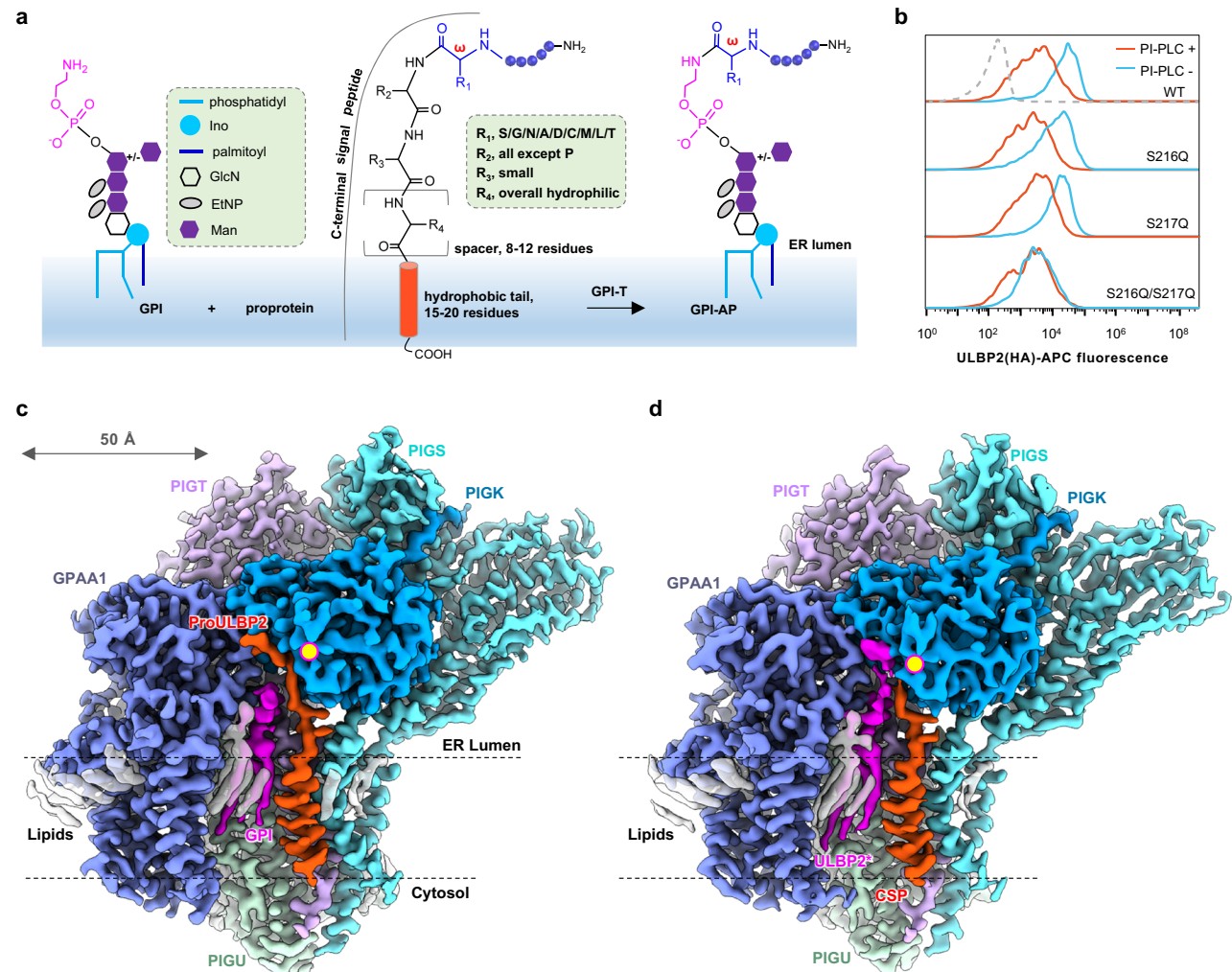

**Fig. 1 | Structures of GPI-T in complex with substrates and products.**
**a** Schematic representation of GPI-T's reaction. The ω-site residues are listed in the box by their occurrence rate[13] from high to low. The preferences for the, ω+1, ω+2, and ω+3 residues (single-letter abbreviations) are indicated in the dashed box. A "+/-" sign indicates the optional Man4 modification for human GPIs. Amino acid residues in the C-terminal signal peptide (CSP) are abbreviated as single letters. ER, endoplasmic reticulum; EtNP, ethanolamine phosphate; GlcN, glucosamine; Ino, inositol; Man, mannose. **b** PI-PLC sensitivity assay of ULBP2 mutants. The staining of HA-tagged wild-type (WT), and single and double glutamine mutants of Ser216/Ser217 on HEK293 cells treated with (red) or without (cyan) PI-PLC were assessed using fluorescently labeled anti-HA antibodies using fluorescence-activated cell sorting. The grey dashed line indicates background staining. See Supplementary Fig. 15a for the gating strategy. PI-PLC, phosphatidylinositol-specific phospholipase C. **c**, **d** Cryo-EM map of GPI-T with substrates **c** and products **d**. The density corresponding to various components is color-coded as indicated. The catalytic dyad residues are indicated by a yellow dot.

detailed catalytic mechanism, and the safeguard measures for catalytic fidelity.

Here, we report protein and cell engineering strategies that allowed the successful purification and subsequent structure determination of the substrate and product complexes of GPI-T. The structures reveal key determinants for substrate recognition and rationalize GPI-T's broad proprotein specificity, and inform a caspase-like catalytic mechanism that involves a fishing rod-like movement of GPI. Moreover, our work suggests a safeguard mechanism against accidental cleavage. In the absence of proproteins, a loop invading the catalytic site keeps the GPI-T complex at an inactive state that is stabilized by hydrogen and ionic interactions. For activation, the auto-inhibitory loop is converted to a crucial part of the catalytic cleft by a drastic conformational change that replaces the aforementioned interactions with repulsive interactions. This energetically unfavorable process is proposed to be powered by proprotein binding, particularly at its topologically characteristic membrane-insertion domain, thus preventing accidental activation. Our work rationalizes how the GPI-T complex recognize proproteins with broad specificity and how it integrates individually poorly conserved CSP features to achieve catalytic fidelity.

## Results

### Characterization of a proprotein substrate

The human GPI-T complex can process over 150 different proprotein substrates. To capture GPI-T with a proprotein, selecting a proprotein with a relatively higher affinity for the enzyme was desirable. Considering this, we identified the UL16 binding protein 2 (ULBP2) as a suitable candidate because it was previously co-purified with GPI-T[26,27]. ULBP2 is a major histocompatibility complex class I-related GPI-AP that activates natural killer cells through binding with the NKG2D receptor[28]. In a previous study[29], Ser216 of ULBP2 was assigned as the ω-site, with Ser217 as a possible alternative. To determine the exact ω-site, we conducted further investigation through mutagenesis.

To facilitate mutagenesis, we constructed a plasmid for the recombinant expression of ULBP2. Additionally, we introduced a hemagglutinin (HA)-tag for detecting ULBP2 surface expression by fluorescence-activated cell sorting (FACS) (Supplementary Fig. 2). While most GPI-APs depend on GPI-anchoring for surface display, and FACS signals are lost with an incompetent ω-site, proULBP2 is known to anchor to the cell surface via its CSP without GPI-anchoring[30], as also demonstrated by the high FACS signal of proULBP2-expressing PIGK-knockout (KO) cells (Supplementary Fig. 3a).

To distinguish between the two types of anchoring, we modified the basic FACS assay by incorporating a step involving phosphatidylinositol-specific phospholipase C (PI-PLC) treatment[31]. PI-PLC, a bacteria toxin, cleaves the diacylglycerol moiety of mature GPI-APs, releasing them from the membrane and causing a loss of FACS signal. In contrast, surface proULBP2 lacks the GPI anchor and thus remains resistant to PI-PLC. Therefore, the PI-PLC sensitivity of ULBP2 surface staining can be used to assess ω-residue competency.

We generated two single and one double glutamine mutant of ULBP2 Ser216 and Ser217, as the bulk glutamine sidechain is known to block GPI-T activity[1–4]. Interestingly, the double mutant ULBP2 S216Q/S217Q, but not the single mutants, showed resistance to PI-PLC (Fig. 1b), indicating that ULBP2 contains two ω-sites, as seen in other GPI-APs[13].

### Protein and cell engineering enable structure determination of GPI-T with substrates and products

To ensure the proprotein's integrity, a dead GPI-T mutant[32] (PIGK C206S) which still binds proproteins[27] was co-expressed with His-tagged proULBP2 (Supplementary Fig. 2) in PIGK knockout (KO) HEK293 cells. The inactive enzyme (GPI-T^C206S) and proULBP2 were then co-purified by tandem affinity chromatography using a Strep-tag

on the PIGU subunit of the GPI-T complex and a His-tag on proULBP2. The relative yield of the second affinity chromatography over the first was approximately 40%, suggesting a relatively tight substrate-bound complex. Consistently, gel filtration (Supplementary Fig. 3b) and SDS-PAGE (Supplementary Fig. 3c) showed co-elution of the Michaelis complex (GPI-T^sub) as a symmetric and monodisperse peak. This purification scheme yielded 0.5 mg of GPI-T^sub per liter of culture.

To capture an enzyme-product complex, we hypothesized that impairing downstream GPI-AP maturation (Supplementary Fig. 1a) could enhance the transamidase's ability to efficiently hold its product(s), thereby enabling the co-purification of the enzyme-product(s) complex. To test this hypothesis, we genetically preserved the inositol acyl chain (Supplementary Fig. 1b), which is likely required for efficient binding with GPI-T, by disrupting *PGAP1*, a gene encoding a GPI-AP deacylase[33]. This disruption is known to affect downstream vesicle transport and GPI-AP remodeling[34], which, in turn, would detain ULBP2 in the ER membrane and further facilitate enzyme-product binding.

Initial attempts to purify the co-expressed GPI-T and ULBP2 using the same affinity purification strategies as GPI-T^sub yielded an insufficient amount (60 μg per liter of culture) of the product complex, mainly due to a low yield (4%) during the second affinity chromatography step. The low yield suggested weaker binding of the products to the enzyme compared with substrates. We reasoned that an artificial proprotein with a stable core, such as the thermostable green fluorescence protein (TGP)[35], may express at a higher level than ULBP2 and thus promote the formation of the enzyme-product complex. Moreover, a fluorescent GPI-AP would allow for convenient assessment of its stoichiometry to the TGP-tagged GPI-T subunits through in-gel fluorescence[35]. Therefore, we constructed a fluorescent proprotein called proULBP2* by grafting the N-/C-terminal signal peptide and the ω−9 region of proULBP2 onto TGP[35] (Supplementary Fig. 2). The PI-PLC sensitivity assay confirmed successful GPI-anchoring of the chimera protein (Supplementary Fig. 3d). Subsequently, we co-expressed and co-purified proULBP2* with the wild-type GPI-T. As expected, the yield for the enzyme-ULBP2* complex (dubbed GPI-T^prod) increased by 3.1-fold compared with the enzyme-ULBP2 complex. Gel filtration fractions (Supplementary Fig. 3e) showing approximately equimolar amounts of GPI-T subunits and ULBP2* on in-gel fluorescence (Supplementary Fig. 3f) were used for structural analysis.

We used single-particle cryo-electron microscopy (cryo-EM) to determine the structures of GPI-T^sub and GPI-T^prod to 3.22-Å and 2.85-Å resolutions, respectively (Fig. 1c, d, Supplementary Figs. 4, 5, Supplementary Table 1). The cryo-EM map of GPI-T^sub was sufficiently clear to build 2,358 GPI-T residues (93% completion), along with 34 lipid/detergent molecules, 3 glycosylation sites, 4 disulfide bonds, and 2 metal ions. In addition to the enzyme, the extra density allowed model building for proULBP2, covering the entire CSP and five residues N-terminal to the ω-site (Fig. 2a). The rest of the proprotein was disordered in the model, consistent with previous[36] and present findings (Supplementary Fig. 3d) that the CSP and the ω−9 region are sufficient for GPI-T recognition. Importantly, a blob of density allowed the building a GPI substrate near proULBP2 (Fig. 1c).

For GPI-T^prod, consistent with the purification results (Supplementary Fig. 3f), densities supporting a GPI-AP with the ω, ω−1, and ω−2 residues were evident. Additionally, the CSP, the other product, was visualized in the structure (Fig. 1d).

### Unexpected importance of the sole PIGT TMH in proprotein binding

ProULBP2 interacts with GPI-T^C206S through a large, buried surface area of 1,821 Å² that covers a 65-Å long footprint (Supplementary Fig. 6a). It inserts into the membrane via an almost traversing α-helix (Figs. 1c, 2a). Introducing helix-breaking proline residues[37] or charged residues[37,38] into the hydrophobic helix, and shortening the helix[16,39] result in reduced or abolished GPI-T activity.

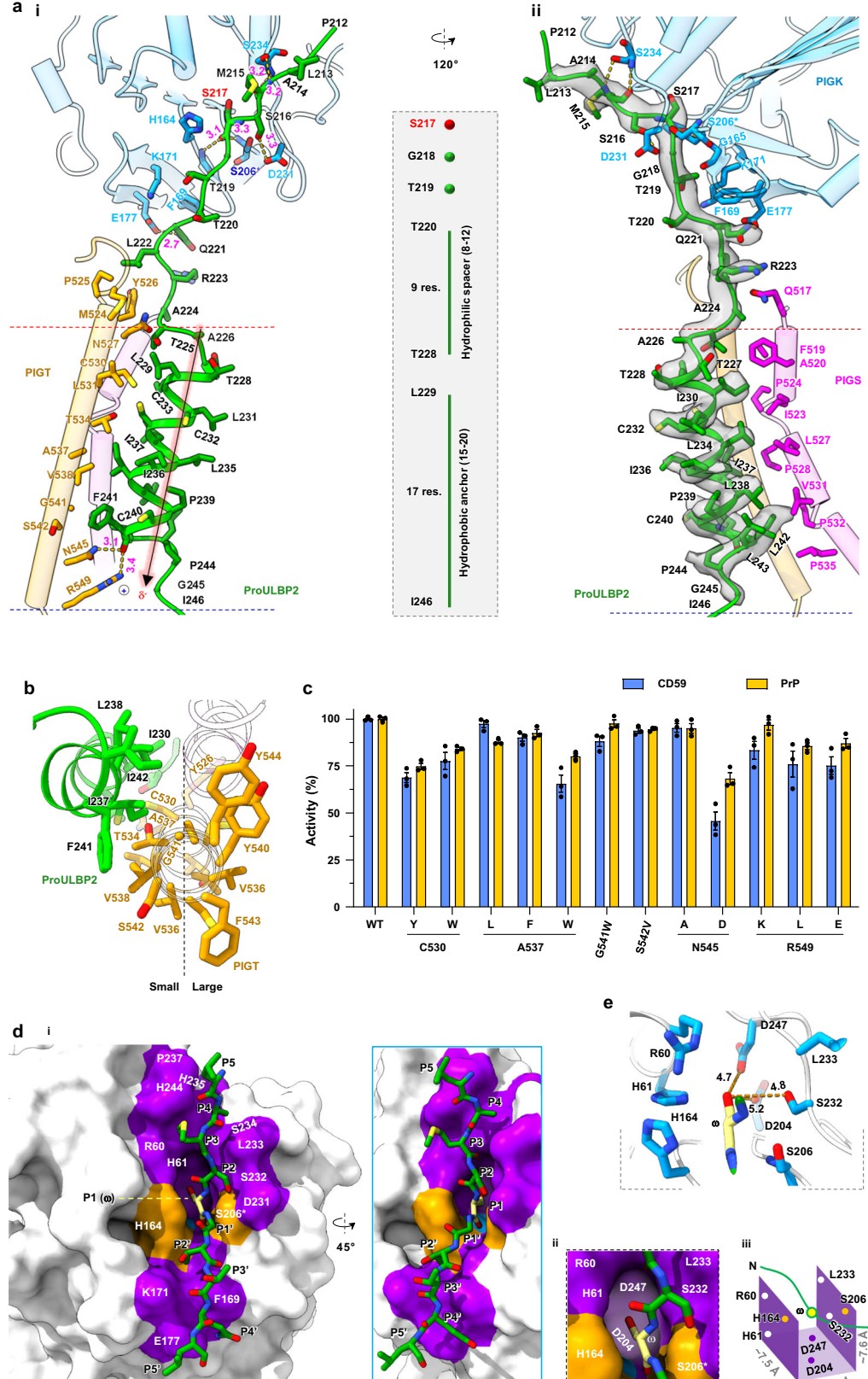

The helix interacts with the transmembrane helix (TMH) 2 of PIGS and the sole TMH of PIGT (Fig. 2a). While a systematic scanning mutagenesis of the 18 residues in PIGS TMH2 did not reveal any functionally important residues (Supplementary Fig. 6b), mutagenesis identified important structural features for CSP binding in PIGT. Specifically, the CSP-interacting face of the PIGT TMH is composed of

relative small residues compared to the opposite face (Fig. 2b), creating a relatively flat surface proposely to accommodate the varying shapes of CSPs from diverse GPI-APs.

To test functional relevance of the structural observation, we created mutations and tested their apparent activity using the aforementioned FACS assay with two endogenous GPI-AP markers: CD59, a

**Fig. 2 | Structural basis for GPI-T's broad proprotein specificity. a** Interactions between proULBP2 (green cartoon) and GPI-T$^{C206S}$ (cylinder and cartoon, color-coded by subunits as indicated). Side chains of GPI-T$^{C206S}$ are shown for proULBP2-binding residues with a cut-off of 3.3 Å for H-bonding and 5.0 Å for hydrophobic interactions. H-bonds are depicted by dash lines with distances in Å (i). The cryo-EM density of proULBP2 is shown as a transparent grey surface (ii). The black arrow indicates the N-to-C direction (positive to negative dipole ends) of the transmembrane helix (TMH). The dashed box shows the pattern of the CSP. An asterisk on S206 of GPI-T$^{C206S}$ indicates the dead mutant. **b** The uneven distribution of side-chain sizes in the PIGT TMH (orange). The TMH of proULBP2 is colored green. **c** Apparent activity of PIGT mutants. Surface staining of two GPI-AP markers (CD59,

blue; PrP, orange) in PIGT knockout cells expressing indicated mutants was assessed using fluorescence-activated cell sorting. Activity relative to the wild-type (WT) is plotted as mean ± s.e.m. ($n$ = 3 independent experiments). See Supplementary Fig. 15b for gating strategy. Source data are provided as a Source Data file. **d** Architecture of subsites (purple surface) that hold proULBP2 residues (green stick) adjacent to the ω-site (yellow). The mutated catalytic residues (His164 and Ser206) are colored orange. An overview (i), an expanded view (ii), and a schematic drawing (iii) of the S1 pocket are shown. Residues N- and C-terminal to the ω-site (P1) are sequentially labeled as P2-P5 and P1'-P5' respectively. **e** Expanded view of the ω-site in the S1 pocket in stick representations. Distance (Å) of hydroxyl on Ser217 to the S1 residues are indicated.

membrane complement regulator that inhibits the formation of the membrane attack complex, and prion protein (PrP), a glycoprotein that causes prion diseases when misfolded. Consistent with the structural data, mutations such as C530Y and C530W reduced the staining of CD59 and PrP, to 70–80% (relative to the wild-type). Mutating Ala537 to leucine had little effects on the staining of CD59 but reduced PrP staining by ~10%. Increasing the bulkiness by the A537F/W mutations further reduced CD59 staining by 9.9%, and 34.4%, respectively (Fig. 2c). This trend was also observed for the PrP marker (A537F, 7.5%; A537W, 19.9%) (Fig. 2c). Finally, although the PrP staining was largely unaffected (Fig. 2c), mutations G541W and S542V reduced CD59 staining by ~10%.

Furthermore, PIGT binds proULBP2 through intramembrane H-bonds involving Asn545/Arg549 with CSP Phe241 and electrostatic attraction between Arg549 and the negative end of the CSP helix dipole (Fig. 2a), both of which are considered strong interactions due to the low dielectric environment of the membrane. Although the N545A mutation had little impact on GPI-T activity, introducing a carbonyl-repulsive mutation, N545D, decreased surface staining by approximately 50% for CD59 and 30% for PrP. The conservative R549K mutation had a minor effect on PrP staining but caused a reduction in CD59 anchoring of approximately 15%. Finally, disrupting the electrostatic interaction by introducing R549L or R549E resulted in a 15%-25% decrease in staining (Fig. 2c). This may explain previous results[39] that efficient GPI anchoring requires a narrow range of CSP hydrophobicity. While short hydrophobic length would affect proper membrane insertion and weaken hydrophobic interactions, a membrane-crossing helix, on the other hand, would extend its negative dipole out of reach for the important Arg549 (Fig. 2a).

Taken together, these results provide a rationale for the key characteristics of the hydrophobic region in CSP, with the crucial corresponding element in the enzyme being the TMH of PIGT. This finding is surprising, as the sole TMH has been reasonably believed[40] to serve as a membrane anchor for PIGT.

### Structural basis for broad proprotein specificity

The selectivity of GPI-T is largely attributed to the ω-site (also known as P1 in the Schechter and Berge nomenclature[41] for proteases) (Fig. 1a). To achieve this, a pocket, and in fact the only deep pocket in the subsites, is used to restrict ω residues. Measuring 3.4 Å × 7.5 Å × 7.6 Å in size, this S1 pocket is composed of residues known to be functionally important[24,25,32,42], with Arg60/His61/His164 and Leu233/Ser232/Ser206 on the two sides, and Asp204/Asp247 at the bottom (Fig. 2d, e). Its shape and charge properties define ω-residue preferences, ranging from high to low occurrences of Ser/Gly/Asn/Ala/Asp/Cys/Met/Leu/Thr[13]. Glycine, alanine, and cysteine are expected to fit well in S1 because they are smaller or similar in size to serine. The relatively deep pocket (Figs. 2d, 2e) can accommodate Asn and Asp, as well as leucine and methionine to a lesser extent due to hydrophobic-hydrophilic mismatch. Aspartate is less preferred than asparagine, probably due to electrorepulsion with Asp204/Asp247. Finally, the narrow opening of the pocket restricts Cβ branching residues. Thus, threonine is a poor ω-site[13],

and the ω-site can be a leucine but not the smaller Cβ-branched valine.

Other features of the CSP element include a non-proline at the ω + 1 site (P1') and a small residue at the ω + 2 (P2') (Fig. 1a). A rigid proline at P1' may impede the insertion of P1 into S1, while a bulky residue at P2' may clash with the catalytic dyad residue His164 (Fig. 2a, d).

The rest of the CSP element and the ω-minus region have low sequence conservation[1]. In accord with this, the GPI-T complex hosts the mainchains of these residues (P5-P5' excepting P1) in a shallow groove while directing their sidechains to the bulk solvent (Fig. 2d). This arrangement would minimize steric clashes with varying side-chains from residuses flanking the ω-site, making the subsite an accommodating architecture. In addition, this region lacks electrostatic and H-bonding interactions (Fig. 2a) which are usually associated with high specificity, resulting in low sequence conservation. Furthermore, the hydrophilic spacer region (Arg223-Ala226) is located in a spacious juxtamembrane chamber without significant enzyme interactions (Fig. 2a). Finally, the membrane-insertion domain is expected to have low sequence conservation because hydrophobic interactions are generally less specific. Overall, the structural findings rationalize the specificity and promiscuity of the CSP element in the proprotein substrate.

### GPI forms a rich network of interaction with GPI-T

The GPI substrate is positioned adjacent to proULBP2 in a functionally significant manner, with the acyl chains inserting in the membrane and the EtNP-modified glycan core approaching the catalytic dyad (Fig. 3a). The positioning of GPI is similar to our previous proprotein-free GPI-T structure[24] (7WLD, referred to as GPI-T$^{apo}$ hereafter although it contains GPI). However, despite the lower resolution of 3.22 Å for GPI-T$^{sub}$, the density allows the visualization of a fully functional GPI with three Man/EtNP residues (Fig. 3a) compared to only one Man/EtNP in the 2.53 Å-resolution GPI-T$^{apo}$[24]. This suggests positive cooperativity for substrate binding.

It is worth noting that, despite the well-defined density for the glycans and EtNPs, there was no apparent evidence for a fourth mannose (Man4). This observation is consistent with previous findings[43] that the mammal GPIs rarely contain Man4. However, it is important to highlight that GPI-T can still accommodate Man4, as the Man3 2-hydroxyl where Man4 may be added points to the bulk solvents (Fig. 3a).

The GPI spans a distance of approximately 44 Å and is enclosed by digitonin and lipid molecules (Fig. 1c). The extensive hydrophobic interactions, hydrogen bonding, and metal coordination (Figs. 3a, 3b) enable the GPI to bind with the enzyme. Two TMHs of PIGU makes the most interactions in the membrane, while several hydrophilic residues from PIGU/PIGT at the juxtamembrane form H-bonds with GPI, as previously observed[24]. Intriguingly, with information from the current study, it was found that instead of binding directly, the mannoses cling to the enzyme through the EtNPs. EtNP1 interacts with GPAA1 His354 through metal coordination (modeled as Mg$^{2+}$) (Fig. 3a). EtNP2 is fixed by GPAA1 Gln355 and Ser51 (Fig. 3a), and when the latter is mutated to

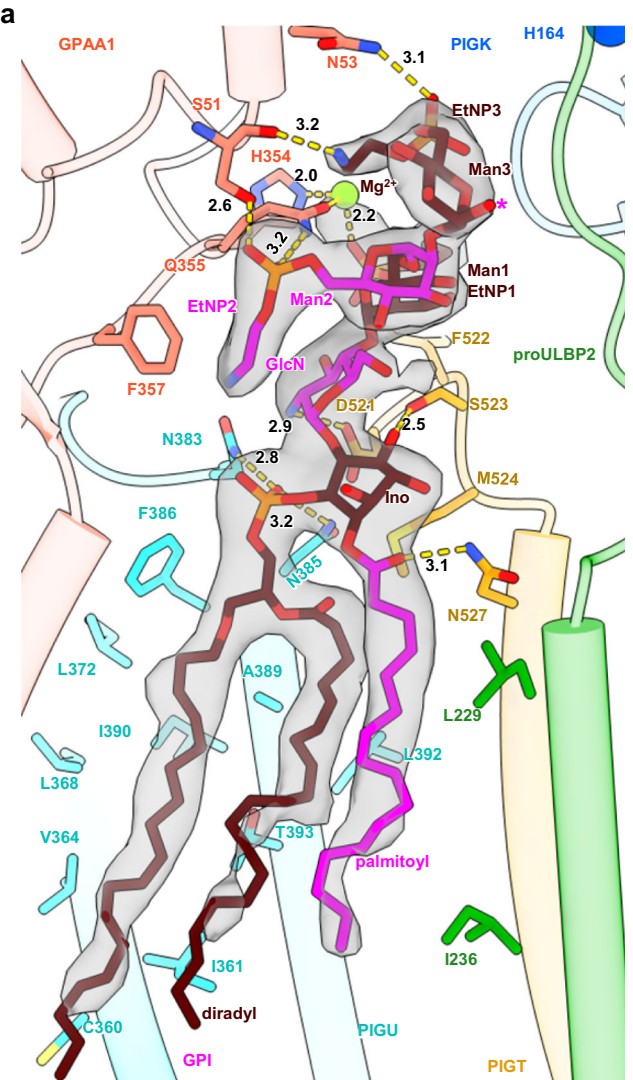

**Fig. 3 | GPI binds GPI-T with a rich network of interactions. a** Interactions between GPI (stick) and GPI-T$^{C206S}$ (cylinder-cartoon with interacting residues in stick representations) (GPAA1, pink; PIGT, orange; PIGU, cyan). The cryo-EM density of GPI is represented by a transparent grey surface. H-bonds/metal coordination with distances in Å are indicated by dash lines. GPI is colored alternatively for better visualization. The catalytic dyad residue His164 (blue sphere) and proULBP2 (green cartoon) are shown for orientation purposes. An asterisk on Man3 indicates the 2-hydroxyl where a fourth mannose may be added. **b** Simplified LigPlot[72] view of the GPI-enzyme interactions. Hydrophobic interactions are indicated by eye slashes and H-bonds/metal coordinations are indicated by dashed lines. GPI is colored alternatively and protein subunits are shaded differently as indicated. GPI-T sub-units and proULBP2 are color-coded as in **a**. EtNP, ethanolamine phosphate; GlcN, glucosamine; Ino, inositol; Man, mannose. **c** Apparent activity of GPAA1 Q355P. Surface staining of the two GPI-AP markers (CD59, left; PrP, right) in GPAA1 knockout cells expressing Q355P (red) or the wild-type GPAA1 (black) using fluorescence-activated cell sorting. Typical results from three independent experiments are shown. See Supplementary Fig. 15b for the gating strategy. Source data are provided as a Source Data file.

leucine, it results in neurodevelopmental disorders and a reduction in GPI-APs level[44]. Finally, EtNP3 interacts with GPAA1 Ser51 and Asn53 (Figs. 3a, 3b). The EtNP-mediated interaction mode explains why effective GPI anchoring necessitates prior EtNP modifications[45,46] in addition to the bridging EtNP3 although they may also be required for other enzymatic steps (Supplementary Fig. 1a).

In line with the multivalent nature of GPI binding, extensive mutations of GPAA1 Tyr49/Ser51/Asn53 did not significantly reduce GPI-T activity (Supplementary Fig. 6c), while the Q355P mutation, intended to eliminate multiple interactions in the vicinity (His354/Gln355/Phe357), resulted in a drastic ~60% reduction in CD59 staining and almost complete abolishment of PrP surface anchoring (Fig. 3c). These findings validate the functional importance of the GPI-binding mode observed in the structure and support the notion that the collective strength of multivalent interaction contributes to strong GPI-binding.

## A fishing rod-like mechanism for GPI attachment

The overall structure of GPI-T$^{prod}$ is very similar to GPI-T$^{sub}$, with a root mean square deviation (RMSD) of 0.40 Å (Supplementary Fig. 7). As expected, the main difference lies in the catalytic site. Despite having a higher resolution than GPI-T$^{sub}$, the ligand density in GPI-T$^{prod}$ near the catalytic dyad was broken between the ω- and ω+1 residues, indicating the production of CSP. This, together with the purification results (Supplementary Fig. 3f), supported the modeling of the GPI-AP product ULBP2* (Fig. 4a), although it should be noted that the density for EtNP3 was less resolved than the rest of GPI (Fig. 4a).

Based on the GPI-T$^{sub}$ and GPI-T$^{prod}$, we propose that GPI-T catalysis follows a general two-phase mechanism similar with cysteine proteases. In the acyl-enzyme phase, the ω-site inserts into the S1 pocket (Fig. 2d, e and 4b), with its carbonyl sandwiched by the two oxyanion hole amines from Cys206 (Ser206 in GPI-T$^{sub}$) and Gly165 (Fig. 4b). Analogous to caspases[47] and legumains[48], this configuration

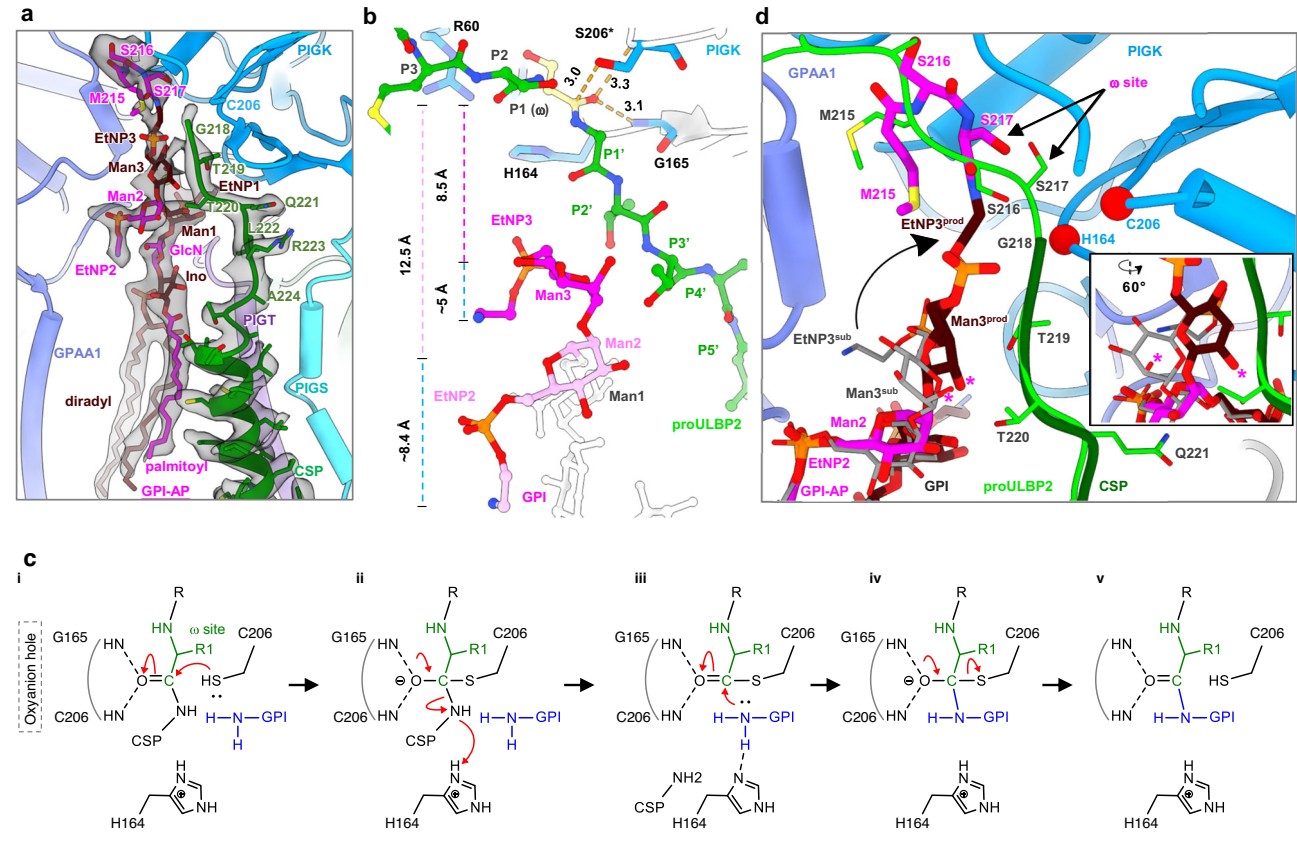

**Fig. 4 | A fishing rod-like mechanism for GPI attachment. a** Interactions between the product ULBP2* and GPI-T (GPAA1, light blue; PIGK, marine blue; PIGT, purple; PGIS, cyan). Cryo-EM density for ULBP2* (alternating colors) and the C-terminal signal peptide (CSP, green) is shown as a transparent grey surface. **b** An expanded view of the substrates (GPI, magenta and grey; proULBP2, green) and PIGK (grey cartoon and blue stick) near the mutated catalytic site of GPI-T$^{sub}$ including the oxyanion hole residues Gly165 and Cys206 (labeled as Ser206* to reflect the C206S mutant), the S1 site residues, and the catalytic dyad (His164 and Cys206). ProULBP2 residues and the Man3/EtNP3 moiety of GPI are shown as stick-and-ball representations. Distances (Å) discussed in the text are indicated as dash lines. **c** Proposed mechanism for GPI-T catalysis. **d** A fishing rod-like movement of GPI from the position in GPI-T$^{sub}$ to the superimposed position in GPI-T$^{prod}$. The subunits (GPAA1, light blue; PIGK, marine blue), ULBP2* (alternating magenta and brown), CSP (dark green) are taken from GPI-T$^{prod}$. ProULBP2 (bright green) and GPI (grey) are superposed from GPI-T$^{sub}$. The Cα atoms of the catalytic dyad are shown as sphere (red). An asterisk on Man3 indicates the 2-hydroxyl where a fourth mannose may be added. The inset illustrates the movement of Man3 from the state in GPI-T$^{sub}$ to that in GPI-T$^{prod}$, viewed at a different angle than the main figure. EtNP, ethanolamine phosphate; GlcN, glucosamine; Ino, inositol; Man, mannose.

further polarizes the C = O bond and facilitates the nucleophilic attack by Cys206 (Fig. 4c, Step i) which is 3.0 Å away from the carbon atom (distance obtained on the Ser206 replacement) (Fig. 4b). As a result, an enzyme-substrate thioester bond is formed at the expense of the collapse of the carbonyl. His164 then acts as a general acid, releasing the CSP, regenerating the carbonyl, and forming the acyl-enzyme intermediate (Fig. 4c, Step ii). In the GPI attachment phase, EtNP3 attacks the intermediate, replacing the CSP. Notably, in the GPI-T$^{sub}$ structure, the reactive EtNP3 is 12 Å away from the scissile bond. To reach the catalytic site, this moiety may undergo a "fishing rod-like" movement, as illustrated by the superimposed GPI-T$^{prod}$ (Fig. 4d), allowing its nucleophilic attack on the carbonyl carbon which is once more polarized by the oxyanion hole (Step iii). Through a new cycle of collapse and reformation of the carbonyl group, GPI attaches to the ω-residue (Step iv), forming the product and freeing the enzyme (Step v).

The 2-hydroxyl of Man3, where Man4 is infrequently added in mammal GPIs[43], undergoes an approximately 180° flip during the "fishing-rod" movement (we acknowledge that the accuracy of Man3/EtNP3 is affected by the less-defined density in GPI-T$^{prod}$). This flipping motion brings Man4 from an open space to a cleft between PIGK and GPAA1 (Fig. 4d), potentially causing clashes. These clashes could be part of the mechanism responsible for the infrequent occurrence of Man4 in mammal GPI-APs. On the other hand, it is also plausible that

steric hindrance may not be an issue due to the flexibility of the mannoses/EtNPs and the spacious local environment.

Although GPI isoforms lacking EtNP3 still bind GPI-T[49], EtNP3 has long been believed to be the sole physiological linker until a recent study[14] demonstrated EtNP2 as an alternative and even the preferred linker in the cases of 5'-nucleotidase Ecto and NetrinG2. In our structure, EtNP2 was located further away from the catalytic dyad than EtNP3 (Fig. 4b). Nevertheless, a fishing rod-like movement, like that observed with EtNP3 (Fig. 4d), around the C6 of Man1 would, with some flexibility, bring EtNP2 in striking distance for the transamidation reaction (~ 4 Å, Fig. 4b). The structural determinants for the proprotein-dependent preference for the bridging EtNP remains an intriguing question for future investigation.

### An autoinhibitory loop regulates GPI-T activity

Cellular protease activities are tightly controlled by mechanisms such as those that involve multi-level regulated zymogens[47], strict acidity requirements[50], and high sequence specificity[51]. Despite its broad specificity, there has been no evidence for a GPI-T zymogen, leading to speculations that the apo enzyme assumes a latent conformation. Indeed, superimposing proULBP2 onto PIGK$^{apo}$ reveals an autoinhibited state. Specifically, the loop containing residues 231-237 (dubbed 231-Loop) invades the active site and blocks the entrance of

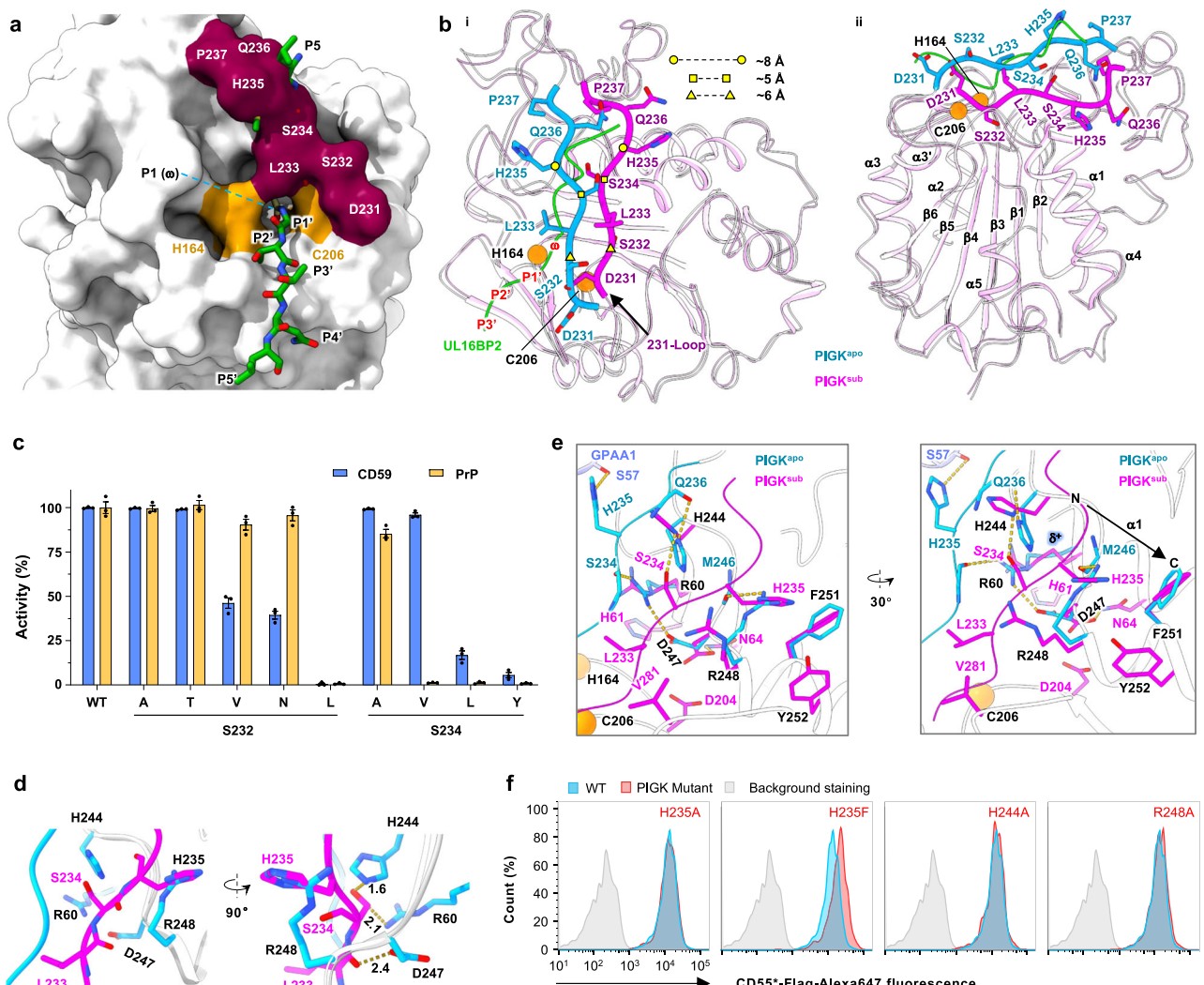

**Fig. 5 | Activation of the GPI-T complex requires an energetically unfavorable conformational change of an autoinhibitory loop. a** Superposing proULBP2 (green stick) onto the apo PIGK structure (surface). The 231-Loop is colored red and the catalytic dyad is colored orange. **b** Structural comparison of PIGK between GPI-T$^{apo}$ (blue for 231-Loop and white for the rest) and GPI-T$^{sub}$ (magenta) viewed at two different angles (i and ii). A part of proULBP2 (green) and the Cα of the catalytic dyad (orange) are shown for orientation purposes. Residues exhibiting significant conformational changes are marked by cycle, square, and triangle symbols. Major secondary structural elements are labeled in ii. **c** Apparent activity of PIGK mutants. Surface staining of the GPI-AP markers (CD59, blue; PrP, orange) in PIGK knockout cells expressing indicated mutants was assessed using fluorescence-activated cell sorting (FACS) with the gating strategy illustrated in Supplementary Fig. 15b. Activity (relative to the wild-type) is plotted as mean ± s.e.m. (n = 3 independent experiments). Source data are provided as a Source Data file. **d** The clash between the 231-Loop from GPI-T$^{sub}$ (magenta) with neighboring residues superimposed

from GPI-T$^{apo}$ (cyan). **e** Energetically unfavorable rearrangement of four sidechains (R60/H244/D247/R248) associated with the conformational change of the 231-Loop. The inactive state (cyan) is stabilized by H-bonds, ionic locks, and cation-π interactions, which are broken upon activation (magenta) and replaced with unfavorable interactions such as electrorepulsion. The N-to-C direction of α1 is indicated by a black arrow, and the positive end of the helix dipole is labeled as δ$^{+}$. The Cα atoms of the catalytic dyad are shown as sphere representation (orange) for orientation purposes. **f** Apparent activity of GPI-T mutants containing substitutions of PIGK residues implicated in the auto-inhibition mechanism. The surface expression of a chimera GPI-AP (CD55*, Supplementary Fig. 2) in PIGK-KO cells transfected with the wild-type PIGK (cyan), PIGK mutants (red), or an irrelevant membrane protein (grey) was assessed by FACS with gating strategy illustrated in Supplementary Fig. 15c. Typical results of three independent experiments are shown.

subsites 1-5 (Fig. 5a). To reconstruct the active state, as in PIGK$^{sub}$, the autoinhibitory loop undergoes a drastic conformational change. During this process, Ser232, an evolutionarily conserved residue (Supplementary Fig. 8a), undergoes a flip of about 180°, causing the Cα to move by approximately 6 Å (Fig. 5b). While Ser232 tolerated mutations to alanine and threonine, introducing valine or asparagine mutations that were intended to cause incomplete flipping by hydrophobic-hydrophilic mismatch (S232V) or clashing (S232N) with Asp204/Asp247 (Supplementary Fig. 8b) resulted in a ~60% reduction CD59 staining (Fig. 5c). Further clashes by S232L completely abolished activity for both CD59 and PrP (Fig. 5c), despite the wild-type-like expression level (Supplementary Fig. 9). Similarly, the side chains of

Ser234 and His235 also undergo a ~180° flip. Ser234 tolerated the alanine mutation, suggesting its hydroxyl is not necessary for shaping the subsite cleft. Although mutating it to valine had no effect on CD59 staining, it abolished the anchoring of PrP (Fig. 5c), suggesting different proprotein sensitivity for S234V. Further increasing the sidechain volume by S234L or S234Y reduced CD59 staining to 16.8% (S234L) and 5.6% (S234Y).

Notably, the four consecutive residues in the 231-Loop from Asp231 to Ser234 form a crucial part of the subsite cleft including the all-important S1 (Fig. 2d). Therefore, the activation requires the auto-inhibitory loop to not only flip out but also to rearrange with three-dimensional accuracy. This double insurance mechanism would

prevent accidental activation caused by random conformational walks of the loop, serving a counter measure for GPI-T's promiscuity.

## GPI-T activation involves energetically unfavorable conformational changes in PIGK

Even though the flipping of the 231-Loop does not cause noticeable backbone changes in the vicinity, several side chains, including Arg60, His244, Asp247, and Arg248, rearrange themselves to avoid steric clashes (Fig. 5d, Supplementary Movie 1). This rearrangement not only breaks several inter- and intra-subunit hydrogen/ionic bonds, but also introduces energetically unfavorable interactions. Specifically, Arg60$^{apo}$ is stabilized by salt bridging with Asp247$^{apo}$ and H-bonding with Ser234$^{apo}$. However, in PIGK$^{sub}$, it is pushed toward His61 and the positive end of the helix dipole of α1, causing electrorepulsion (Fig. 5e). Similarly, Asp247, the other component of this ionic lock, moves towards the electrorepulsive Asp204$^{sub}$, although the nearby Asn64 may offer some compensation (Fig. 5e). Furthermore, Arg248$^{apo}$ is stabilized by a H-bond with Met246$^{apo}$ and a cation-π interaction with Phe251$^{apo}$. The transformation brings this charged residue close to two hydrophobic residues, Leu233 and Val281. Finally, the rearrangement breaks an inter-subunit H-bond (GPAA1 Ser57$^{apo}$ with PIGK His235$^{apo}$) and replaces it with an intra-subunit H-bond between His244$^{sub}$ and Ser234$^{sub}$ (Fig. 5e, Supplementary Movie 1). Taken together, the inactive PIGK$^{apo}$ state is locally stabilized while the active PIGK$^{sub}$ state is locally destabilized. This contrast would set an energetic barrier to further suppress leak activity, providing yet another safeguard regulation.

To further test the auto-inhibition model, we designed three mutations to destabilize the inactive state: PIGK H235A, H244A, and R248A. Additionally, we introduced PIGK H235F to not only destabilize the inactive state by eliminating the H-bond with GPAA1 Ser53 but also stabilize the active state by forming hydrophobic interactions with PIGK Phe251/Tyr252 (Fig. 5e). We anticipate that these mutants would increase the apparent activity and show higher surface staining of GPI-AP markers compared to the wild-type PIGK. However, FACS results with the endogenous marker CD59 did not show differences among the PIGK mutants and the wild-type (Supplementary Fig. 10a). This result was challenging to interpret, as the surface display of CD59 is influenced by enzymes and transporters in the entire GPI-AP biogenesis pathway rather than GPI-T alone. Moreover, it is unclear whether GPI-T is the rate-limiting enzyme in this context. Nevertheless, one straightforward explanation for the lack of differences could be that the limited availability of the endogenous CD59 proprotein becomes a limiting factor for the cell-based FACS assay, thus failing to report the full potential of gain-of-function mutants.

To overcome the substrate availability issue, we modified the FACS assay by introducing an overexpressed GPI-AP reporter. A chimera TGP- and CD55-based GPI-AP (dubbed CD55*, Supplementary Fig. 2) was constructed similarly to the pULBP2$^{chimera}$ used in GPI-T$^{prod}$. To differentiate GPI-AP-expressing cells (GFP fluorescence) from PIGK-expressing cells, we added a mCherry tag to PIGK. GPI-T activity was assessed in PIGK-KO cells by fluorescence gating for CD55-TGP expression, PIGK expression, and the surface display of CD55* (via Flag-tag). Although the apparent GPI-T activity for PIGK H235A, H244A, and R248A were similar to that of the wild-type, cells transfected with PIGK H235F exhibited higher fluorescence intensity (Fig. 5f, Supplementary Fig. 10b). This result is consistent with the abovementioned double-action design for PIGK H235F. The extent to which other mutants also promote GPI-T activity remains to be investigated using more quantitative and, preferably, test-tube biochemical assays.

## Proprotein binding triggers dramatic and concerted subunit movements that power GPI-T activation

Next, we explored the driving force that may overcome the abovementioned energy barrier during activation. We compared GPI-T$^{apo}$

with GPI-T$^{sub}$ beyond PIGK. Aligning the two structures using the PIGK luminal domain as the reference reveals overall movements by as large as Cα displacement of 12 Å for the other subunits (Fig. 6a). Interestingly, these movements, alongside that of GPI, seem to be concerted and directed towards the elongated substrate-binding site (Supplementary Movie 2). Moreover, they occur primarily at a global level, as relatively small differences are observed when subunits are aligned individually (Supplementary Fig. 11).

Noticeable local conformational shifts include those in the two THMs and the juxtamembrane region of PIGS (Supplementary Fig. 12a, 12b). Specifically, proprotein binding induces movement of the two TMHs towards the lumen by roughly half a helix turn. Further, the juxtamembrane loop preceding TMH2 flips and shifts towards PIGK (Supplementary Fig. 12a), obstructing the channel between PIGS and PIGK that was recently[52] suggested as a potential pocket for CSP. The most striking transformation was a ~180° flip of Tyr512, resulting in a Cα displacement of ~7 Å.

However, extensive mutagenesis of these regions (Supplementary Fig. 12c), and indeed the two TMHs (Supplementary Fig. 6b), including a total of 20 alanine scanning and glycine/proline mutations that were intended to change mainchain flexibility and affect the conformational changes, did not affect GPI-T activity in the cell-based assay, indicating that the conformational changes are functionally inconsequential. In line with our findings, a previous study[53] of the yeast PIGS (Gpi17p) with eight mutations in these regions also did not compromise GPI-T activity. Therefore, the local conformational changes are likely the result of the CSP-binding rather than being part of the conformational driving force for GPI-T activation, although the possibility of these changes making a collective contribution to activation cannot be ruled out.

The likely inconsequential changes mentioned above suggests that the large rigidbody-type subunit movements (Fig. 6a, Supplementary Movie 1) are responsible for GPI-T activation. We propose that the CSP, especially its hydrophobic portion, plays a major role in initiating the rigid movements. Our proposal is based on several observations. First, GPI-binding alone is insufficient to lift the auto-inhibition, as evidenced by the inactive state of the GPI-bound GPI-T$^{apo}$ structure[24,25]. Second, while proULBP2 binding activates GPI-T, the interactions from the ω-minus residues are minor (Fig. 2a). Similarly, the uneven distribution of these interactions is more pronounced in GPI-T$^{prod}$. ULBP2*, which has only two moderately ordered ω-minus residues, would have a minor influence on enzyme-product interactions. In addition, the less defined densities suggest that the protein residues and EtNP3 are about to detach from PIGK, further weakening its role in stabilizing the active state. Conversely, the CSP remains ordered in the structure (Fig. 4a). Finally, the attempting proposal of the hydrophobic region in CSP making a major contribution was made because the GPI-T movements are overall larger in the transmembrane region than the lumen domain (Fig. 6a), and so are the degree of orderliness caused by the CSP (Supplementary Fig. 13). In addition, the narrow range of CSP hydrophobicity[39] and the somewhat fragility of interactions at the membrane domain[16,37–39] (Fig. 2c) suggest an important role of the hydrophobic region.

In summary, we propose the following model for the GPI-T activation cycle (Fig. 6b). In the absence of proproteins, GPI-T assumes an inactive state (Fig. 6b i). The binding of the 30-residue long CSP particularly at the membrane region generates a considerable amount of energy that stabilizes neighboring elements in the membrane and enables rigid movements of the subunits (Fig. 6b ii). This process flips the autoinhibitory loop out of the catalytic site. Meanwhile, CSP residues near the ω-site induces a competent cleft for catalysis. GPI-T remains in an active state until CSP is released (Fig. 6b iii), following which GPI-T reverts to the inhibited state (Fig. 6b iv) due to energetically unfavorable interactions (Fig. 5e) to prevent potential off-target proteolysis. In doing so, GPI-T integrates individually poorly conserved

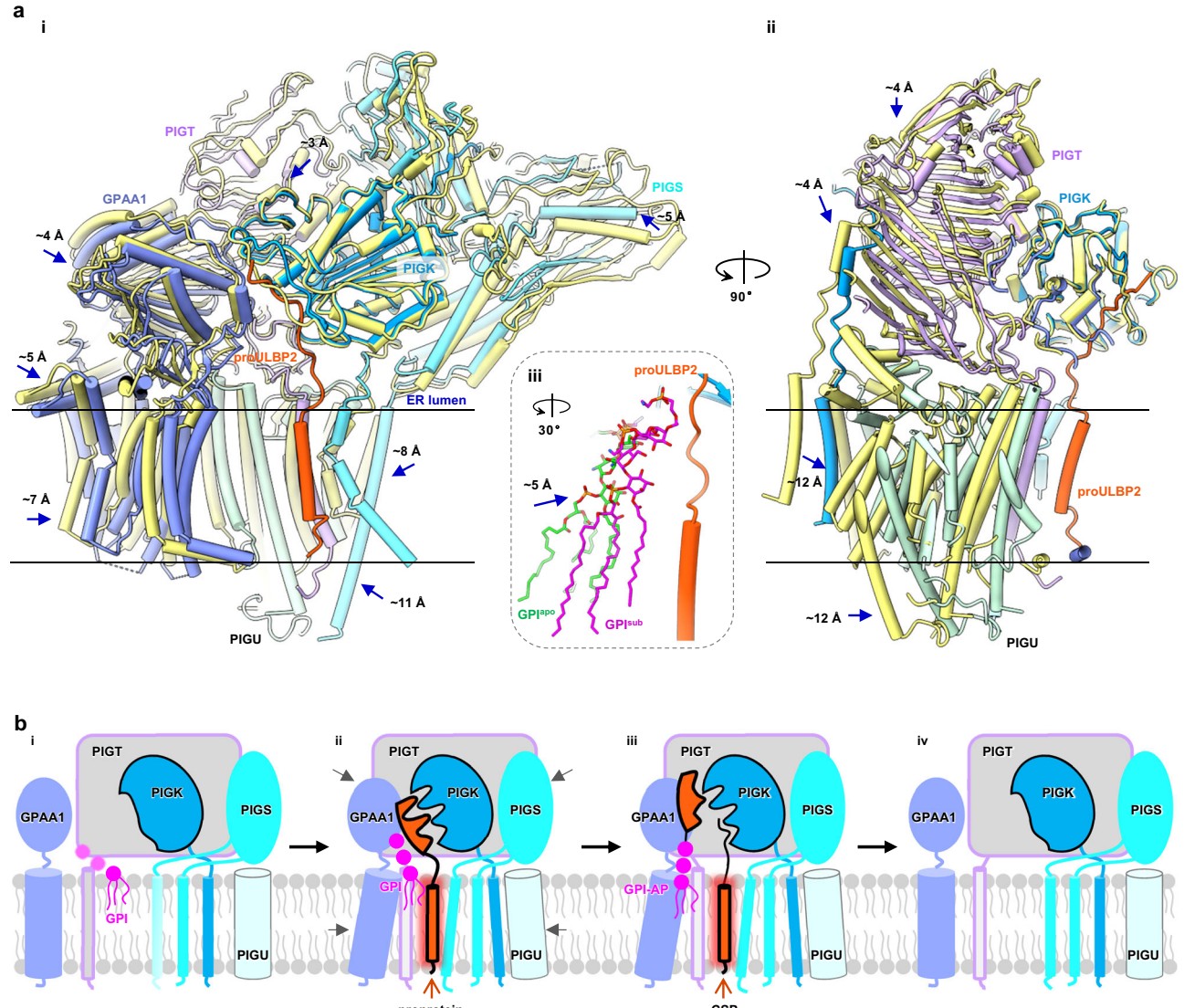

**Fig. 6 | Proposed mechanism for GPI-T activation. a** Conformational differences between GPI-T[apo] (yellow) and GPI-T[sub] (GPAA1, light blue; PIGK, marine blue; PIGT, purple; PIGS, cyan; PIGU, pale cyan) viewed at two different angles (i, ii). Inset (iii) illustrates the movement of GPI from the position in the GPI-T[apo] structure (GPI[apo], green) to that in the GPI-T[sub] structure (GPI[sub], magenta). The superposed proULBP2 is colored red. Blue arrows indicate regions with significant conformational changes. **b** A scheme of the activation cycle. In the absence of proprotein substrates (i),

GPI-T (color-coded as indicated) exists in an inactive state with a disrupted catalytic site. The faded coloring of GPI (magenta) and the PIGS TMH2 (cyan) indicate their relatively high flexibility. The binding of proprotein substrates (red), particularly the CSP region (glow), to GPI-T triggers concerted conformational changes that alleviate the inhibition by the 231-Loop. Meanwhile, proprotein residues near the ω-site induce the formation of a catalytic cleft (ii) for catalysis (iii), following which the products are released and GPI-T reverts to the inactive state (iv).

CSP features to make a collectively strong selectivity filter for its substrates to prevent unintentional cleavage.

## Discussion

The human GPI-T complex catalyzes the essential GPI attachment step for the biosynthesis of over 150 GPI-APs. Despite its importance, the mechanism by which the CSP region, lacking an apparent sequence consensus, controls GPI-T activity and how it maintains a balance between broad substrate specificity and fidelity has been a long-standing mystery. In this study, we present the structures of the Michaelis complex of GPI-T, as well as the enzyme-products complex. These structures reveal that the previously reported proprotein-free GPI-T structures[24,25] exist in an inactive state, while the current structures represent the active state. Through mutagenesis and structural analyses, we demonstrate that seemingly weak features of the CSP region collectively form a strong selective filter for the activation of GPI-T, elucidating how this region controls substrate suitability despite

the low sequence consensus. Furthermore, the architecture of the proprotein-binding site rationalizes GPI-T's broad substrate specificity and its moderate selectivity of ω-site residues. The structures also suggest caspase-like catalytic mechanisms for substrate activation and catalysis. Moreover, the atomic details of the subsites provide a precise framework for in-sillico prediction of GPI-APs and their ω-sites and help to boost the accuracy of existing algorithms[54–57].

The surface expression of two GPI-AP reporters in the cell-based assay responded to some GPI-T mutants with large differences despite the overall similar trend. For mutations of the proprotein-binding sites, this is not unexpected because the two GPI-APs use different ω-minus and CSP sequences. However, such large differences were not anticipated for the GPI-binding site mutant GPAA1 Q355P (Fig. 3c) because GPI is the common substrate for all GPI-APs. This discrepancy may be explained by the recent discovery[14] that the GPI anchoring of CD59 can be realized through EtNP2 in addition to EtNP3. Because EtNP2 needs to move away from Gln355 and towards the catalytic dyad in

EtNP2-mediated but not in EtNP3-mediated modifications, Q355P is expected to impact less for CD59 than PrP assuming the latter is attached to GPI through EtNP3.

The successful co-purification of GPI-T with both products opens up exciting possibilities for future structural studies of other GPI-AP-processing enzymes. However, it raises important concerns about product inhibition, especially considering GPI-T's role in processing numerous proproteins. Nevertheless, protein inhibition in native cells is likely to occur at a reduced level compared to the overexpression system used in the study. Moreover, in native cells, nascent GPI-APs undergo PGAP1-mediated inositol-deacylation, which is essential for efficient ER export. This remodeling process should weaken the interaction between GPI-APs and GPI-T, facilitating product release. Further, due to the lower yield of GPI-T$^{prod}$ compared to GPI-T$^{sub}$, it is plausible that proprotein substrates bind more tightly to GPI-T than the products and thus outcompete the products to assist product release.

GPI-T$^{prod}$ was purified using tandem affinity chromatography with tags on ULBP2* and GPI-T. Interestingly, the tag-free CSP was tightly bound to GPI-T during co-purification, as no distinct classes of ULBP2*-only particles were observed in the cryo-EM data processing. This observation suggests that CSP is either co-released or released after ULBP2*. Supporting this notion, the density for ULBP2* in GPI-T$^{prod}$ appears less defined than CSP, indicating a more dynamic and departing conformation for ULBP2*.

The autoinhibition mechanism, rather than zymogen as for caspases[58] or strict sequence requirement as for thrombin[51], may be more suitable for GPI-T. Unlike signaling caspases that only emerges in particular cellular events, GPI-T needs to standby, ready to process its myriad substrates (hence no strict sequence conservation) that are required at all cellular stages. Therefore, switching reversibly between the active and inactive states by a self-loop would be significantly more biologically efficient than the zymogen approach.

Given the structural similarity between PIGK and caspases, it is worthy to discuss the similarity and differences between their activation mechanisms. The autoinhibitory loop of PIGK is at a structurally similar region of the 341-loop[47] of caspases which regulates substrate binding and catalytic activities (Supplementary Fig. 14a, 14b). However, the conformational change and its driving force are distinctly different. In zymogenic caspase-7, an elbow loop proceeding the 341-loop is pushed towards the direction of the catalytic dyad by the intersubunit linker at the dimerization interface[59]. As a result, the 341-loop assumes an 'elevated' conformation, distorting the active site (Supplementary Fig. 14c). The cleavage of the linker allows the elbow and the 341-loop to relax back and to form the active site[58]. The nearby 381-loop also changes conformation to facilitate the activation (Supplementary Fig. 14d). In the GPI-T complex, the corresponding elbow loop and the 278-loop (the equivalent of the caspase 381-loop) do not show noticeable changes between the active and inactive states (Supplementary Fig. 14e). Instead, the autoinhibitory loop undergoes a flip and 'downward' motion (Fig. 5b, Supplementary Fig. 14e, 14f). Finally, in the GPI-T complex, the elbow loop is protruding to the bulk solvent (Supplementary Fig. 14g) and is thus unlikely subjected to being pushed by other subunits. Therefore, the two enzymes appear to use an evolutionarily convergent structural element for autoinhibition but divergent mechanisms for activation.

## Methods

### Molecular cloning and constructs

The genes encoding the five human GPI-T subunits had been cloned in our previous study[24]. Briefly, the genes encoding GPAA1 (Genbank ID NP_003792.1), PIGK (NP_005473.1), PIGS (NP_149975.1), and PIGT (NP_057021.2) were amplified from cDNA clones provided by the authors' institute. The gene encoding PIGU (NP_536724.1) was PCR amplified in-house using overlapping oligonucleotides. The PCR products were

Gibson assembled[60] (Cat. EG21202S, BestEnzymes Biotech, Lianyungang, China) into various versions of the pBTSG vector (Addgene #159420) that carries DNA encoding a thermostable green fluorescence protein (TGP)[35]. The TGP tag has been shown to be compatible with GPI-T activity in our previous study[24]. The affinity tag at the C-terminal of TGP were the following for the purification of the proprotein-bound GPI-T$^{C206S}$: GPAA1, 2×Flag; PIGK$^{C206S}$, hemagglutinin (HA); PIGS, Myc; PIGT, no tag; PIGU, Strep II. The affinity tags for the purification of the products-bound GPI-T$^{prod}$ were the same as that for GPI-T$^{C206S}$ except that (1) a His-tag was fused with PIGT; and (2) no tag was fused with PIGU.

For experiments where the expression of a TGP-based GPI-AP needs to be distinguished from the expression of PIGK (Fig. 5f, Supplementary Fig. 10b), the TGP-tag of PIGK was replaced with a mCherry tag using Gibson assembly. The mCherry tag has the same sequence as that of Uniprot #A0A366VY15 except for the following four mutations that extends its fluorescence lifetime: W148S, I166V, Q168Y, and I202R (W143S, I161V, Q163Y and I197R in Ref. 61).

Three expression plasmids for ULBP2 were used in this study. For the flow cytometry experiment, the construct (dubbed pULBP2$^{FACS}$, Supplementary Fig. 2) was designed to have an N-terminal hemagglutinin (HA) tag (YPYDVPDYA) for surface staining purposes. To facilitate the gating of ULBP2-expressing cells during flow cytometry, a blue fluorescence protein (TagBFP, residues 2084–2316 of NCBI #MN019124.1) was also expressed by this construct via an internal ribosome entry site (IRES) (nucleotides 2900–3486 of NCBI #MN542793.1) linker. This design aims to exclude non-transfected cells (no TagBFP signal) during flow cytometry analysis. The pULBP2$^{FACS}$ plasmid was constructed as follows. First, a DNA fragment encoding residues 1-26 of ULBP2 (NCBI ID NP_079493.1), the HA tag with flanking Gly-Ser at both ends, and residues 27–246 of ULBP2, was obtained by overlap PCR using chemically synthesized oligos and cDNA clones provided by the authors' institute. A second DNA fragment for the IRES was in-house synthesize by PCR using overlapping oligo nucleotides. A third DNA fragment encoding TagBFP was amplified from the plasmid pLL313-plenti-BFP-T2A-bla (lab collection). The three DNA fragments were Gibson assemblied with the first and the third sandwiching the second into a modified pBTSG that was made by deleting the DNA fragments encoding the His-tagged TGP. After proper processing by GPI-T, pULBP2$^{FACS}$ (Supplementary Fig. 2) would produce an HA-tagged ULBP2 on the surface of TagBFP$^+$ cells. To investigate the ω-site of ULBP2, single mutants S216Q, S217Q, and the double mutant S216Q/S217Q (ULBP2 numbering) were made by standard site-directed mutagenesis using pULBP2$^{FACS}$ as the template.

The second ULBP2 construct, named pULBP2$^{Purif}$ (Supplementary Fig. 2), was designed to co-express with the dead enzyme (GPI-T$^{C206S}$) for purification of the proprotein-enzyme complex. The coding sequence of ULBP2 was cloned into the abovementioned pBTSG variant by Gibson assembly. A DNA fragment encoding an octa-histidine tag flanked by a Gly-Ser linker on both sides was inserted into the coding DNA of ULBP2 such that the Gly-Ser-flanked His-tag is located at the immediate C-terminal of the N-terminal signal peptide (residue 1-26). This construct is expected to produce an N-terminally His-tagged proULBP2 after processing by signal peptidase in GPI-T-defective cells.

The third ULBP2 construct, named as pULBP2$^{chimera}$ (Supplementary Fig. 2), was designed to produce a TGP-containing GPI-AP after GPI-T processing. The construct is cloned into the abovementioned pBTSG variant and expresses the following elements from the N- to the C-terminus: The N-terminal signal peptide of ULBP2 (residues 1–26), TGP with a flanking Gly-Ser linker at both ends, Strep II tag, and a fragment containing the ω−9 residue to the end of ULBP2 (residues 208–246).

As a control for the GPI-anchoring of the recombinant ULBP2 (pULBP2$^{FACS}$), a similar construct was made for CD59 (dubbed pCD59$^{FACS}$, Supplementary Fig. 2). It followed the same design strategy

as pULBP2$^{FACS}$. Therefore, the processed CD59 would contain a HA-tag at its N-terminus after being displayed on the cell surface.

To overexpress a fluorescent GPI-AP marker for the activity assay of potential gain-of-function PIGK mutants (Fig. 5f, Supplementary Fig. 10b), a chimera CD55-TGP constructs was made (dubbed pCD55$^{chimera}$, Supplementary Fig. 2) similarly to pULBP2$^{chimera}$. Specifically, the construct is cloned into the abovementioned pBTSG variant and expresses the following elements from the N- to the C-terminus: The N-terminal signal peptide of CD55 (residues 1-34 of Uniprot #P08174 [https://www.uniprot.org/uniprotkb/P08174/entry]), TGP with a flanking Gly-Ser linker at both ends, a Flag tag, and a fragment containing the ω−9 residue to the end of CD55 (residues 348-381) (Supplementary Fig. 2).

To overexpress and purify phosphatidylinositol phospholipase C (PI-PLC), the encoding sequence of the C-terminal 298 amino acids of *Bacillus cereus* phospholipase C (Genbank ID AAA22665.1 [https://www.ncbi.nlm.nih.gov/protein/AAA22665.1]) with the *Bsp*QI restriction site at both DNA ends was synthesized and subcloned into pUC57 by Azenda (Suzhou, China). The fragment was cut from the pUC57-PIPLC plasmid by *Bsp*QI digestion, gel-purified, and ligated into the *Bsp*QI-digested vector pSb-init[62] using T4 ligase. The resulting plasmid contains a signal peptide for periplasmic expression in *Escherichia coli* and encodes a C-terminally His-tagged PI-PLC.

Mutants of the GPI-T subunits were made using standard PCR-based mutagenesis. All the constructs and mutants were verified by Sanger sequencing.

## Expression and Purification of GPI-T$^{C206S}$ with proULBP2

To prevent GPI-T processing of proULBP2, an HEK293T cell line with the *PIGK* gene knocked out (PIGK-KO) from our previous study[24] was used. To adapt this adherent cell line for suspension, cells were transferred into 30 mL sera-free medium (Cat. 1000, Union, Shanghai, China). After two days of culturing at 37 °C in a CO$_2$ (5%) incubator shaking at 125 r.p.m., cell passaging was performed. The clustered cells became dispersed after seven passages with viability of 95% according to Trypan Blue staining.

To co-express GPI-T$^{C206S}$ and proULBP2, suspension PIGK-KO cells adapted above were transfected with five plasmids for the five GPI-T subunits, and one for proULBP2, as follows. The day before transfection, 1 L cells were diluted to a density of $1 \times 10^6$ mL$^{-1}$ and cultured at 37 °C in a CO$_2$ shaking incubator. A total of 1.5 mg plasmids (mass ratio of PIGS:PIGT:GPAA1:PIGU:PIGK-C206S:pULBP2$^{Purif}$ (Supplementary Fig. 2), 38:21:27.5:42:21:15) and 3 mg polyethylenimine (PEI) were mixed with 50 mL of medium in two separate tubes for 3 min before being pooled together for incubation at room temperature (RT, 20-22 °C) for 20 min. The mixture was then added into 1 L of cell culture which typically had a density of $2 \times 10^6$ mL$^{-1}$. In addition, sodium valproate (Cat. P4543, Sigma) was supplemented at a final concentration of 2 mM to improve protein expression. Cells were harvested after 48 h, washed with PBS buffer, snap-frozen with liquid nitrogen and stored at −80 °C before use.

The complex of GPI-T$^{C206S}$ and proULBP2 was purified using tandem affinity chromatography and gel filtration. All the purification procedures were conducted at 4 °C. Cells from 5 L of culture were solubilized with Buffer A (150 mM NaCl, 50 mM Tris-HCl pH 8.0) supplemented with protease inhibitor cocktail (Cat.B14001, Bimake) and 1%(w/v) lauryl maltose neopentyl glycol (LMNG) and 0.1%(w/v) cholesteryl hemisuccinate (CHS) for 1 h. Cell debris were removed by centrifuging at 48,000 g for 1 h. The supernatant containing solubilized GPI-T$^{C206S}$ and proULBP2 was collected and mixed with 6 mL of Strep Tactin beads (Cat. SA053100, Smart-lifesciences) pre-equilibrated with Buffer A and stirred gently for 1.5 h. The mixture was then pooled into a gravity column for purification and detergent exchange. The beads were washed with 5 column volume (CV) Wash Buffer 1 (0.01% LMNG, 0.001% CHS and 0.1% digitonin (Cat. D82515,

ABCone) in Buffer B (150 mM NaCl, 20 mM Tris-HCl pH 8.0) and 1.5 CV Wash Buffer 2 (0.2% digitonin in Buffer B). After 30 min of incubation, the beads were further washed sequentially with 1.5 CV Wash Buffer 2 and 3 CV Wash Buffer 3 (0.1% digitonin in Buffer B). The enzyme-protein complex was eluted with 5 mM D-desthiobiotin (Cat. Sc-294239A, Santa Cruz), 0.1% digitonin in Buffer B. Fractions were pooled and incubated with 3 mL of Ni-NTA beads supplemented with 10 mM imidazole for 2 h with mild agitation. The beads were packed into a gravity column, washed with 5 CV Wash Buffer 4 (0.1% digitonin, 10 mM imidazole in Buffer B), and eluted with 300 mM imidazole, 0.1% digitonin in Buffer B. The pooled fractions were concentrated with a 100-kDa cut-off concentrator (Cat. UFC810096, Merck millipore) and further fractioned by size exclusion chromatography (Bio-Rad NGC with software ChromLab 3.3.0.09) using a Superose 6 10/300 GL column (Cat. 29-0915-96, Cytiva) with Wash Buffer 3 as the running buffer. Peak fractions were pooled together and concentrated to 25 mg mL$^{-1}$ for cryo-EM grid preparation. Protein concentration was determined by the absorbance at 280 nm measured using a Nanodrop machine with a theoretical extinction coefficient of 636,337 M$^{-1}$ cm$^{-1}$ assuming an equimolar stoichiometry.

## Generation of PGAP1 knock-out cells for the expression of GPI-T with products

To generate PGAP1 knock out cells line, the endogenous gene encoding PGAP1 was disrupted by CRISPR-Cas9 editing using two pairs of sgRNA oligos: sgEx9_Fwd (5′- CACCGTTCTAGTAAAAGTGTCCAAA-3′) and sgEx9_Rev (5′- AAACTTTGGACACTTTTACTAGAAC-3′); and sgEx10_Fwd (5′- CACCGCTTGAAAATCATAGAAAAAT-3′) and sgEx10_Rev (5′- AAACATTTTTCTATGATTTTCAAGC-3′) which were designed using the online server (http://cistrome.org/SSC/)[63]. The oligos, designed to have sticky ends of Type IIs restriction enzyme *Bbs*I (Cat. R3539S, NEB) after annealing, were dissolved in a buffer containing 0.2 M NaCl, 0.1 mM EDTA, and 10 mM Tris HCl pH 7.5 and mixed to have 10 μM of each in a PCR tube. After heating at 95 °C for 3 min, the oligo pairs were annealed by gradual cooling from 94 °C to 25 °C at 1 °C gradients and an 11-s incubation under each temperature. The annealed mix (1 μL) was ligated into the vector pX330 (50 ng) pre-digested with *Bbs*I using T4 ligase (Cat. EL0011, Thermo Fisher Scientific). The ligation products were transformed into DH5α and positive colonies were identified by Sanger sequencing.

For CRISPR-Cas9 gene editing, 8 μg of the sgRNA-carrying plasmids, 0.16 μg of pMaxGFP, 16 μL of P3000 (Cat. L3000008, Thermo Fisher Scientific) were mixed with 250 μL of Opti-MEM medium (Cat. 31985070, Thermo Fisher Scientific). This mix was incubated with 16 μL of Lipofectamine 3000 and 250 μL of Opti-MEM medium at room temperature (RT, 20-22 °C) for 15 min, before being added dropwise to a 6-cm dish containing HEK293T cells (Cat. CRL-3216, ATCC) with 70−90% confluency. Cells were cultured in a 5% CO$_2$ incubator at 37 °C in a Dulbecco's Modified Eagle Medium (DMEM) supplemented with 10% fetal bovine serum (FBS, Cat. 40130ES76, Yeasen, Shanghai, China). After 24 h, cells were washed with 2 mL of PBS, digested with 0.5 mL of 0.1% trypsin (Cat. 25200056, Thermo Fisher Scientific) for 3 min at 37 °C, and re-suspended in 3 mL of DMEM and 10% FBS. Cells were collected by centrifugation at RT at 300 g for 5 min, washed with 10 mL of PBS, and re-suspended with 0.5 mL PBS for fluorescence assisted cell sorting (FACS) using a BD FACSAria Fusion machine operated under software BD FACS Diva (version 8.0.3). The top 5% green fluorescence protein (GFP)-positive cells (~40,000) were collected, serially diluted using DMEM supplemented with 10% FBS, and seeded into 96-well plates. Wells containing single colonies were selected under a microscope, and cells were populated for 10-12 d in a stationary CO$_2$ incubator. Cells were detached using trypsin, resuspended in 200 μL of DMEM and 10% FBS, and divided into two aliquots (160 μL and 70 μL) for scaling-up and PCR-identification, respectively. Positive clones which were expected to produce a DNA fragment of

approximately 1 kb (2.5 kb for wild-type cells) were screened by PCR using the following primer pair 5′- ATTTCCCTATGATGATGCTGGT −3′ and 5′- GCCAATGGAAAACAAAATTCCCTT −3′. The genomic deletions were further verified by DNA sequencing of the PCR products.

## Purification of GPI-T with products

For the expression of the active GPI-T with its protein substrate, the PGAP1-KO cell line was used. The cells were adapted using the same procedure as outlined for the PIGK-KO cell line for suspension culturing.

Eight liters of cells expressing GPI-T and proULBP2* (Supplementary Fig. 2) were collected by centrifugation, rinsed once with PBS buffer, and re-suspended in Buffer A. Purification was performed at 4 °C. Cells were added with 1%(w/v) LMNG and 0.1%(w/v) CHS for solubilization for 1 h. Cell debris were removed by centrifugation at 48,000 g for 1 h. The supernatant was gently stirred with 8 mL of Strep Tactin beads (Strep tag is on ULBP2*). The beads were packed into a gravity column, washed with 5 CV of 0.01% LMNG, 0.001% CHS and 0.1% Digitonin before being incubated with 0.2% digitonin in Buffer A for 20 min. The beads were then washed with 1.5 CV of 0.2% digitonin and 3 CV of 0.1% digitonin in Buffer A, and the proteins were eluted with 5 mM D-desthiobiotin, 0.1% digitonin in Buffer A. The elution was incubated with 3.5 mL pre-equilibrated Ni-NTA beads for 2 h, after which time the beads were packed into a gravity column and washed with 6 CV Buffer A containing 10 mM imidazole and 0.1% digitonin. After an elution step with 250 mM imidazole and 0.1% digitonin in Buffer A, the samples were concentrated with a 100-kDa cut-off concentrator and further fractioned on a Superose 6 10/300 GL column with 0.1% digitonin in Buffer A as the running buffer. Fractions were examined by SDS-PAGE and in-gel fluorescence for the integrity of the TGP-tagged GPI-T complex and the presence of the TGP-containing product. Desired fractions were pooled and concentrated to 8 mg mL$^{-1}$ for cryo-EM grid preparation. Protein concentration was determined by the absorbance at 280 nm measured using a Nanodrop machine with a theoretical extinction coefficient of 628,987 M$^{-1}$ cm$^{-1}$ assuming an equimolar stoichiometry. Purified samples were analyzed by SDS-PAGE and the bands were visualized using a portable TGreen Transilluminator (Cat. OSE-470, Tiangen, Beijing, China) for in-gel fluorescence. Gel images were captured using a smartphone.

## Expression and purification of PI-PLC

*Escherichia coli* MC1061 cells from a single colony carrying the plasmid encoding PI-PLC were cultured in Terrific Broth (TB, 17 mM KH$_2$PO$_4$ and 72 mM K$_2$HPO$_4$, 1.2%(w/v) tryptone, 2.4%(w/v) yeast extract, 0.5% (v/v) glycerol) supplemented with 25 mg L$^{-1}$ chloramphenicol at 37 °C for 16 h. The overnight culture was 1:100 seeded into fresh TB medium. After culturing at 37 °C for 2 h, the growing temperature was shifted to 22 °C and cells were cultured for another 1.5 h before induction with 0.2% arabinose for 16 h. Cells from 1 L of culture were harvested by centrifugation at 4,000 g for 20 min, resuspended in 20 mL TES buffer (0.5 M sucrose, 0.5 mM EDTA, and 0.2 M Tris-HCl pH 8.0), and stirred at 4 °C for 30 min for dehydration. Cells were lysed through abrupt rehydration by the addition of 40 mL ice-cold water and stirring at 4 °C for 1 h to release the periplasmic extracts. The sample was centrifuged at 20,000 g for 30 min. The supernatant containing PI-PLC was collected, adjusted to contain 150 mM NaCl, 2 mM MgCl$_2$ and 20 mM imidazole, mixed with 0.5 mL pre-equilibrated Ni-NTA resins, and agitated gently at 4 °C for 2 h. The beads were packed into a gravity column, washed with 20 CV of 30 mM imidazole in the buffer containing 150 mM NaCl, 20 mM HEPES pH 7.5 and eluted with 300 mM imidazole in the same buffer. Fractions were pooled, aliquoted, snap frozen with liquid nitrogen, and stored at −80 °C before use. Protein concentration was calculated using absorbance at 280 nm with the theoretical extinction coefficient of 67,091 M$^{-1}$ cm$^{-1}$.

## Flow cytometry analysis for surface staining of the native GPI-AP reporters CD59 and PrP

Wild-type or GPI-T single-subunit KO HEK293 cells[24] were maintained in DMEM supplemented with 10% fetal bovine serum (FBS, Cat. 40130ES76, Yeasen) in a 24-well plate inside a CO$_2$ stationary incubator at 37 °C. For transfection, 1 μL of Lipofectamine 3000 was incubated with 25 μL Opti-MEM medium at RT for 5 min. The mix was added to a separate mix containing 0.5 μg plasmid, 1 μL of P3000 (Cat. L3000008, Thermo Fisher Scientific) and 25 μL of Opti-MEM medium. After incubation at RT for 15 min, the mixture was added dropwise to the cells for transfection. Two days after transfection, cells were washed with PBS and digested with 0.12 mL of 0.1% trypsin for 2 min at 37 °C before being re-suspended in 0.72 mL of DMEM and 10% FBS to saturate trypsin. Cells were then harvested by centrifugation at RT at 800 g for 3 min, washed with 0.8 mL of PBS, and resuspended in 0.1-0.5 mL PBS.

For the surface staining of the GPI-AP reporters, phycoerythrin (PE)-labeled CD59 antibody (Cat. 12-0596-42, clone OV9A2, Thermo Fisher Scientific) or PrP antibody (Cat. 12-9230-42, clone 4D5, Thermo Fisher Scientific) were used as a 500-fold (CD59) or 50-fold (PrP) dilution for incubation with the cells for 15 min at dark. Cells were washed once with PBS and resuspended in 0.3 mL of PBS for flow cytometry (Beckman CytoFlex LX with software CytExpert 2.4.0.28) monitored at two wavelength pairs (488/525 for GFP, 561/585 for PE). Cells (typically 40,000) were gated using the GFP channel (from the expression of TGP-tagged single GPI-T subunits) and analyzed for positive signal for the PE channel (for surface staining of CD59/PrP) using the software FlowJo (version v10.0.7, BD Life Sciences) (Supplementary Fig. 15b). For the apparent activity of GPI-T mutants, the percentage of CD59/PrP immune staining in KO-cells transfected with the mutant was normalized to a negative control (cells expressing the TGP-tagged Patched, an unrelated membrane protein) and the positive control (cells expressing the TGP-tagged wild-type protein). The expression and integrity of all mutants were separately confirmed by SDS-PAGE in-gel fluorescence. Data reported in this work were from three independent experiments except for mutants with no effect on GPI-T activity. Data for flow cytometry are included in Source Data.

All the commercial antibodies used in this study were validated by the manufacturers (see Report Summary for details).

## Flow cytometry analysis for surface staining of the HA-tagged GPI-AP reporters

For surface staining of HA-tagged ULBP2, HEK293 cells or PIGK-KO HEK293 cells transfected with pULBP2$^{FACS}$ (Supplementary Fig. 2) were treated the same way as mentioned in the section above. To the resuspended cells, Alexa Fluor 647 conjugated HA-Tag antibody (Cat. 3444 S, clone 6E2, Cell Signaling Technology, 1:100 dilution) was incubated with the cells for 15 min in the dark. Cells were rinsed with PBS and resuspended in -0.3 mL of PBS for flow cytometry (Beckman CytoFlex LX). Cells (typically 60,000) were gated using the TagBFP channel (405/450 nm) and the Alexa Fluor 647 channel (638/660 nm) as reporters for successful transfection and surface expression, respectively (Supplementary Fig. 15a). To distinguish proULBP2 which may be transported to the cell surface and anchored to the plasma membrane by the CSP from ULBP2 which is anchored by GPI, cell staining was also performed with a pretreatment step using PI-PLC. Because PI-PLC selectively releases GPI-anchored ULBP2, the difference of surface staining between untreated and PI-PLC treated samples reflects the level of ULBP2 on the cell surface; the remaining signal after PI-PLC treatment is deemed as surface staining caused by proULBP2. For PI-PLC treatment, 10 μg of home-purified PI-PLC was added to the cells (0.5 mL). The digestion was carried out at 37 °C for 1.5 h. Cells were washed with PBS before being stained using Alexa Fluor 647-labeled anti-HA antibody.

For surface staining of HA-tagged CD59 (Supplementary Fig. 2), the procedure was the same for HA-tagged ULBP2 except that only the PIGK KO cell line was used. The same anti-HA antibodies were used as for HA-tagged ULBP2 instead of the anti-CD59 antibodies.

## Flow cytometry analysis for surface staining of ULBP2* (TGP-ULBP2 chimera protein)

For surface staining of ULBP2*, which is a TGP-containing fluorescent GPI-AP, HEK293 cells were transfected with pULBP2$^{chimera}$ (Supplementary Fig. 2) using Lipofectamine 3000. Two days after transfection, cells were washed with PBS, treated with trypsin, rinsed with PBS, and resuspended in 0.5 mL of PBS. Cells were incubated with or without 10 µg of PI-PLC for 1.5 h at 37 °C. Cells were washed with PBS, and incubated with 10 µg of a TGP-specific nanobody (Sb44)[35]. After 15 min, unbound Sb44 was removed and a PE-conjugated second antibody (Cat. 3739 S, clone 9B11, Cell Signaling Technology, 1:300 dilution) recognizing the Myc-tag on Sb44 was then added for staining at RT for 15 min. After a rinse step, cells were subjected to FACS analysis as outlined above.

## Flow cytometry analysis for surface staining of CD55* (TGP-CD55 chimera protein)

The overexpressed CD55* marker was used for the assay of the potential gain-of-function PIGK mutations. PIGK-KO cells were co-transfected with pCD55$^{chimera}$ (Supplementary Fig. 2) and the plasmid encoding a mCherry-tagged PIGK (wild-type, mutants, or an irrelevant membrane protein-protein) using Lipofectamine 3000. The plasmid ratio for PIGK and CD55* was 1: 1 (wt: wt). The rest of the procedures were the same as above. For surface staining, cells were resuspended in 0.1 mL Alexa Fluor 647 conjugated Flag antibody diluted in PBS (Cat.15009 S, clone D6W5B, Cell Signaling Technology, 1:100) and incubated at RT for 30 min protected from light. Cells were washed once with PBS and resuspended in 0.5 mL of PBS for flow cytometry (Beckman, CytoFlex LX) monitored at three wavelength pairs (488/525 nm for TGP, 561/610 nm for mCherry, 638/660 nm for Alexa Fluor 647). Cells (typically 40,000) were gated using the GFP channel (from expression of CD55*), the mCherry channel (for expression of PIGK mutants or WT) and analyzed for signal for APC channel (for surface staining of Flag-tagged CD55*) using the software FlowJo (BD Life Sciences). The gating strategy can be found in Supplementary Fig. 15c. Data reported in this work were from three independent experiments.

## Cryo-EM data collection

Purified GPI-T$^{C206S}$ in complex with proULBP2 (2.5 µL) at a concentration of 25 mg mL$^{-1}$ was applied onto glow-discharged Quantifoil Au R1.2/1.3 (300 mesh) grids and blotted with filter paper for 3 s with a blotting force of 5 at 4 °C, with 100% humidity in a Vitrobot Mark IV (FEI) chamber before flash plunged into liquid ethane for cryo-cooling. Grids of the ULBP2*-bound GPI-T were prepared using the same procedure as that of GPI-T$^{sub}$ except that the concentration was at 8 mg mL$^{-1}$ and the blotting force was at 7.

Grids were loaded in a Titan Krios G4 cryo-electron microscope (Thermo Fisher) operated at 300 kV with a 70 µm condenser lens aperture, spot size 4, magnification at 130,000 × (corresponding to a calibrated sampling of 0.932 Å per physical pixel), and a Falcon 4i direct electron device equipped with a Selectris X energy filter operated with a 20 eV slit (Thermo Scientific). Movie stacks were collected automatically using the EPU software (Thermo Fisher) with the Falcon 4i detector operating in counting mode at a total exposure time of 3.51 s, yielding 1,080 frames per EER (electron event representation) movie and a total dose of 50 e$^{-}$/Å$^{2}$.

## Cryo-EM data processing

A total of 4555 GPI-T$^{sub}$ movies and 10,379 GPI-T$^{prod}$ movies were collected and processed similarly in RELION (v3.1)[64] and cryoSPARC (v4.2.0)[65]. Each electron-event representation movie of 1080 frames were fractionated into 40 subgroups and beam-induced motion was corrected by RELION's own implementation. Exposure-weighted averages were then imported to cryoSPARC and the contrast transfer function parameters for each micrograph were estimated by CTFFIND4[66]. Particles were blob-picked and extracted with a box size of 270 pixels, and subjected to several rounds of 2D classification and heterogeneous refinement (3D classification) using our previous 2.53 Å human GPI-T map[24] lowpass filtered as reference, to remove contaminants or poor-quality particles. The good particles were then converted for Bayesian polishing in RELION, which was subsequently imported back to cryoSPARC for heterogeneous refinement. The final 3.22 Å GPI-T$^{sub}$ map from 176,889 particles, and 2.85 Å GPI-T$^{prod}$ map from 34,261 particles were obtained by local refinement. The resolution of these maps was estimated internally in cryoSPARC by gold-standard Fourier shell correlation using the 0.143 criterion. Details for data processing are in Supplementary Information (Supplementary Figs. 4 and 5) and Supplementary Table 1.

## Model building and refinement

The models of individual subunits of the previously reported GPI-T (PDB ID 7WLD) were first fitted into the cryo-EM map using Chimera[67]. The GPI-T model was then adjusted using Coot[68] (version 0.9.6) guided by the cryo-EM density. The model of proULBP2 was built ab initio in Coot based on the amino-acid sequence and the cryo-EM density. The cryo-EM density was insufficiently clear to assign accurate acyl-chain length and saturation; GPI was built to contain the 1-alkyl-2-acyl chain for the phosphatidylinositol moiety as it is the most possible composition according to a previous study[69]. Although not experimental verified, a Mg$^{2+}$ ion was modelled into the GPI-binding site based on the coordination distances, and that Mg$^{2+}$ is the most abundant ion in cells. The model containing GPI-T, proULBP2, and other ligands was refined with Phenix. real_space_refine[70] (version 1.19.2-4158), yielding an averaged model–map correlation coefficient (CCmask) of 0.86 (GPI-T$^{sub}$) and 0.81 (GPI-T$^{prod}$). Structures were visualized using UCSF ChimeraX1.1[71] and PyMol (version 2.3.3) (https://pymol.org/2/).

## Reporting summary

Further information on research design is available in the Nature Portfolio Reporting Summary linked to this article.

## Data availability

The coordinates for the model GPI-T$^{sub}$ and GPI-T$^{prod}$ generated in this study have been deposited in the PDB under accession codes 8IMY and 8IMX, respectively. The cryo-EM density maps for the GPI-T$^{sub}$ and GPI-T$^{prod}$ generated in this study have been deposited in the Electron Microscopy Data Bank with accession codes EMD- 35576 and 35575. The coordinates for the previously published model GPI-T$^{apo}$ are fetched from the PDB database under accession code 7WLD. Uncropped images of Supplementary Fig. 3f, and tabular data for Figs. 2c, 3c, 5c, and Supplementary Figs. 6b, 6c and 12c are provided in the Source Data file. Source data are provided in this paper.

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

## Acknowledgements

We thank the staff members at the Core Facilities of Molecular Biology and Cell Biology in Shanghai Institute of Biochemistry and Cell Biology, and the staff members of the Center of Cryo-EM of Fudan University for technical support and assistance. This work has been supported by the National Natural Science Foundation of China (82151215, D.L.; 32171194 & 32371256, Q.Q.; 32201000, T.L.), the Strategic Priority Research Program of CAS (XDB37020204, D.L.), Science and Technology Commission of Shanghai Municipality (22ZR1468300, D.L.), the Shanghai Post-doctoral Excellence Program (2021378, T.L.), the China Postdoctoral Science Foundation (2022M720805, Z.Zhou), and the start-up funds from Shanghai Stomatological Hospital & School of Stomatology, Fudan University (Q.Q.).

## Author contributions

Y.X. and T.L. purified the complex. Y.X. performed functional assays. J.H. created cell lines. Z.Zhou prepared and screened cryo-EM grids and collected cryo-EM data with assistance from Y.C. and Z.Zhu, under the supervision of Q.Q. and Y.Z. Q.Q. and Z.Zhou processed cryo-EM data. D.L., Y.X. and T.L. designed experiments. D.L. and Q.Q. oversaw the project. D.L. wrote the manuscript with input from Q.Q., Y.X. and T.L.

## Competing interests

The authors declare no competing interests.
