## [Peer review file · Nature Communications]

REVIEWER COMMENTS

Reviewer #1 (Remarks to the Author):

This is an impressive manuscript from Xu et. al, reporting the cryo-EM structures of GPI-T, a GPI transamidase, in both substrate and product bound states. Although the cryo-EM structure of GPI bound GPI-T has been determined before, this new structural work provides a large body of additional information for defining how GPI-T recognizes the substrates and how the substrate is transferred to GPI. By comparing this substrate bound structure with the previous apo-structure, the authors also proposed an autoinhibitory mechanism of GPI-T. Overall, the cryo-EM work is of good quality. Both cryo-EM maps were determined at high resolution; thus, the model building for the substrate (proULBP2) and product (GPI modified ULBP2) seems to be reliable. I therefore support the publication of this work at Nature Communications. I have a couple of suggestions that may help to improve the manuscript further:

(1) It is unclear to me how the GPI modified substrate and CSP are released from GPI-T after the catalytic reaction is finished. In the cryo-EM structure of product bound GPI-T, both of GPI modified substrate and CSP still make extensive interaction with GPI-T, and these binding modes are almost identical to that observed in the pre-catalytic state. I am wondering, in the cryo-EM dataset of product bound GPI-T, are there any classes that represent only GPI-modified ULBP2 or CSP bound GPI-T? If there is no such class, can authors provide any explanation for why GPI-modified ULBP2 and CSP still bind GPI-T tightly after the reaction? It would be helpful if the authors can provide more discussion on how the GPI-modified substrate and CSP are released.

(2) The authors propose an autoinhibitory mechanism of GPI-T, in which a short loop of GPI-T (residues 231-237) adopts a conformation that partially blocks the substrate entrance in the apo-state of GPI-T. The substrate binding alters the conformation of this loop that allows the initiation of the catalytic reaction. To test this model, can authors mutate some residues surrounding this loop in the autoinhibit conformation (such as, Arg60)? If the mutation can only affect the autoinhibit conformation of this loop but doesn't disturb its active conformation, such GPI-T mutants should exhibit higher activity.

Reviewer #2 (Remarks to the Author):

GPI anchoring is a highly conserved mechanism in eukaryotic cells, and this post translational modification is critical in various biological processes. The addition of the GPI to the proprotein is carried out by the GPI-T complex of proteins, which consists of five subunits. The GPI-T complex exhibits broad specificity recognizing a broad pattern of C-terminal peptides. Using protein and cell engineering strategies the authors successfully purified and solved the structure of the GPI-T complex in its substrate bound form as well as in its product bound form. To solve the structure of the substrate bound form the authors mutated C206S in PIGK, so that the enzyme complex still binds proproteins, but the protein is catalytically dead. To capture the product bound state the authors genetically engineered a cell line to disrupt the gene encoding GPI-AP deacylase which is downstream of the GPI-T complex.

I overall find the protein and genetic engineering to be cleverly done, and the structural work good and I would recommend publishing the paper.

I have some suggestions for changes to the text and a couple of sections that needs more details or explanation.

Major points:

I think the surface staining experiments are not well described in the text. For example, line 113-115

is assuming the reader already know the assay and if you don't, you need to search in the methods to understand what the assay is, the authors don't even mention that this is fluorescent activated cell sorting in the text.

Also, I would like an introduction to CD59 and prion labeling, these are mentioned and used, but never introduced.

It's unclear to me why the GFP fusion was used? And I would like an illustration of what this construct looks like, I have a hard time understanding the sentence describing the complex in line 147-149.

I would like a little more detail in the description of the purification of the complex, you can find it in the methods, but I think a couple of sentences in the main text would be useful (line 135-138).

Minor points:

I think it would make sense to call it the GPI-T complex throughout the text and not just GPI-T
Line 60-64 is very confusing. I believe the "is" on line 60 should be a "was", and maybe there should be a "is the" between EtNP2 and preferred instead of the "a" on line 61. But the section could use a re-write.

For line 106-109 I find the sentence very hard to read and understand, I would consider a re-write.

Also on line 109-110, Ser216 and Ser217 of what protein?

On line 248 the reference to Fig. 1a does not make sense.

In figure 3c I think there is a typo it says Q35P in the left plot.

First section of the discussion is hard to understand, should it be "committed" in line 503?

On line 504 an "are" is missing between "here" and "the"

On line 509 to 510 it should read: ... and help to boost accuracy of existing algorithms.

Reviewer #3 (Remarks to the Author):

Glycosylphosphatidylinositol transamidase (GPI-T), a complex of five transmembrane proteins, catalyzes posttranslational modification of many eukaryotic cell surface proteins by GPI-anchors. The enzyme recognizes a GPI attachment signal peptide located at the C-terminus of proproteins (CSP) and replaces it with a preassembled GPI that acts as membrane anchors of the modified protein. More than 150 human proteins are GPI-anchored, however, sequences of their CSPs are different and have only several common characteristics. A remarkable feature of GPI-T is its ability to recognize all those only weakly conserved sequences and faithfully attaches GPI. This research group previously reported 3D structure of human GPI-T with associated GPI but lacking proprotein, termed GPI-Tapo. To clarify structural basis of transamidation-mediated GPI attachment, authors took a clever approach and were able to purify two forms of GPI-T, GPI-Tsub bearing two substrates (GPI and a proprotein), and GPI-Tprod bearing two products (a GPI-anchored protein and a CSP). Comparing 3D structures of those three GPI-Ts determined by cryoEM, authors report several important points relevant to understanding the GPI attachment mechanism: (1) CSP makes functionally important association with sole TMD of PIGT.; (2) GPI associates with GPI-T through interactions of lipid moiety with PIGU, interactions of three ethanolamine phosphates with GPAA1, and interactions of juxtamembrane region with PIGT and PIGU.; (3) Presence of a deep pocket accommodating only a small amino acid at the omega site (to which GPI is attached) and shallow grooves connecting to both sides of the deep pocket accounting for accommodation of CSP of various amino acid sequences.; (4) A fishing rod-like mechanism of GPI

attachment to the C-terminus of the omega site; (5) Demonstration of inactive state of GPI-T, in which the catalytic region is covered by an auto-inhibitory loop of PIGK.; (5) Activation of GPI-T by flipping out of the auto-inhibitory loop and its rearrangement to generate environment of the catalytic region.; and (6) Possible mechanism that prevents unintentional cleavage.

Based on these findings, authors proposed a model of GPI attachment reactions. The model is supported by structural data and explains unique features of GPI-T actions. These findings greatly advance understanding of mechanisms of GPI attachment.

The paper is clearly written and is easy to read. I have only minor points and corrections.

Minor points.

1. Fig1a, c and d. GPI is illustrated as having four mannoses in Fig 1a whereas GPI in GPI-Tsub and GPI-Tprod had three mannoses. It would be better if illustration and actual structures are consistent.

2. Fig. 3 and Fig. 4. It would be informative for readers if authors comment on whether fourth mannose can be accommodated in GPI-Tsub without affecting the local structure and whether a fishing-rod-like movement can occur with the fourth mannose.

3. p3, line 44. GPI biosynthesis insufficiency has not been reported in cancers. Amplification and upregulation of genes of GPI-T components were reported in cancers. The sentence needs to be modified for accuracy

Other corrections.

1. p3, line 45. Hepatic not haptic?
2. p3, line 52. Glucosamine not glycosylamine.
3. p4, line 55. Later rather than latter?
4. p20, line 442. PIGS not PGIS.

RESPONSE TO REVIEWER COMMENTS

Reviewer #1

This is an impressive manuscript from Xu et. al, reporting the cryo-EM structures of GPI-T, a GPI transamidase, in both substrate and product bound states. Although the cryo-EM structure of GPI bound GPI-T has been determined before, this new structural work provides a large body of additional information for defining how GPI-T recognizes the substrates and how the substrate is transferred to GPI. By comparing this substrate bound structure with the previous apo-structure, the authors also proposed an autoinhibitory mechanism of GPI-T. Overall, the cryo-EM work is of good quality. Both cryo-EM maps were determined at high resolution; thus, the model building for the substrate (proULBP2) and product (GPI modified ULBP2) seems to be reliable. I therefore support the publication of this work at Nature Communications. I have a couple of suggestions that may help to improve the manuscript further:

We thank the reviewer for the insightful comments and support review of our work.

(1) It is unclear to me how the GPI modified substrate and CSP are released from GPI-T after the catalytic reaction is finished. In the cryo-EM structure of product bound GPI-T, both of GPI modified substrate and CSP still make extensive interaction with GPI-T, and these binding modes are almost identical to that observed in the pre-catalytic state. I am wondering, in the cryo-EM dataset of product bound GPI-T, are there any classes that represent only GPI-modified ULBP2 or CSP bound GPI-T?

We thank the reviewer for raising this point.

During our study, we did not observe distinct GPI-T classes with only GPI-modified ULBP2* (ULBP2*) or CSP bound. While such states may exist, they were not prominent enough in our data processing probably due to the low particle numbers. The limited particles could be attributed to our purification strategy or related to the product-release mechanism.

The GPI-T^{prod} complex was purified by tandem affinity chromatography with affinity tags on the ULBP2* and GPI-T. Even if GPI-T with only CSP bound exists, the purification approach would exclude most of them. Therefore, unless ULBP2* dissociates from GPI-T^{prod} during cryo-EM grid preparation, the amount of CSP-only GPI-T particles should be negligible. However, if GPI-T with only ULBP2* bound were present, it should have been enriched in our sample. The absence of this class, combined with the well-defined CSP density in the cryo-EM map, suggests that the release of CSP either occurs with, or subsequent to, the release of GPI-APs.

If there is no such class, can authors provide any explanation for why GPI-modified ULBP2 and CSP still bind GPI-T tightly after the reaction?

ProULBP2/ULBP2 represents a high-affinity substrate/product with GPI-T, as its endogenous form has been co-purified with the enzyme in previous studies (ref. 27 and 28). Biochemically, the strong interactions between ULBP2*/CSP with GPI-T can be explained by the abundant interactions observed in the GPI-T^{prod} structure.

Physiologically, these tight interactions may raise concerns about potential undesired product inhibition. However, it is likely that in native cells, product inhibition occurs at a minimal level due to several factors:

- 1) The concentration of endogenous GPI-T and products in native cells is expected to be much lower than in the overexpression system used in our study;
- 2) PGAP1, which is absent in our expression system, should facilitate product release by changing the composition (inositol deacylation) and location (ER export) of GPI-APs;
- 3) Considering the lower yield of GPI-T^{prod} compared to GPI-T^{sub}, it is possible that proprotein substrates bind more tightly to GPI-T than the products, thus further assisting in product release.

It would be helpful if the authors can provide more discussion on how the GPI-modified substrate and CSP are released.

Because the density of ULBP2* is less defined than the CSP, and because of the lack of CSP-only GPI-T classes, we propose that CSP is either co-released or released after ULBP2*.

We have added the following text in the Discussion section.

“The successful co-purification of GPI-T with both products opens up exciting possibilities for future structural studies of other GPI-AP-processing enzymes. However, it raises important concerns about product inhibition, especially considering GPI-T’s role in processing numerous proproteins. Nevertheless, protein inhibition in native cells is likely to occur at a reduced level compared to the overexpression system used in the study. Moreover, in native cells, nascent GPI-APs undergo PGAP1-mediated inositol-deacylation, which is essential for efficient ER export. This remodeling process should weaken the interaction between GPI-APs and GPI-T, facilitating product release. Further, due to the lower yield of GPI-T^{prod} compared to GPI-T^{sub}, it is plausible that proprotein substrates bind more tightly to GPI-T than the products and thus outcompete the products to assist product release.

GPI-T^{prod} was purified using tandem affinity chromatography with tags on ULBP2* and GPI-T. Interestingly, the tag-free CSP was tightly bound to GPI-T during co-purification, as no distinct classes of ULBP2*-only particles were observed in the cryo-EM data processing. This observation suggests that CSP is either co-released or released after ULBP2*. Supporting this notion, the density for ULBP2* in GPI-T^{prod} appears less defined than CSP, indicating a more dynamic and departing conformation for ULBP2*.”

(2) The authors propose an autoinhibitory mechanism of GPI-T, in which a short loop of GPI-T (residues 231-237) adopts a conformation that partially blocks the substrate entrance in the apo-state of GPI-T. The substrate binding alters the conformation of this loop that allows the initiation of the catalytic reaction. To test this model, can authors mutate some residues surrounding this loop in the autoinhibit conformation

(such as, Arg60)? If the mutation can only affect the autoinhibit conformation of this loop but doesn't disturb its active conformation, such GPI-T mutants should exhibit higher activity.

We thank the reviewer for suggesting this insightful hypothesis-testing experiment.

We selected several residues implicated in stabilizing the inactive conformation for the mutagenesis study. They include PIGK H235A, H235F, H244A, and R248A [PIGK Arg60 was previously identified to be involved in substrate-binding (also in **Fig. 2d** of this study), and our earlier study demonstrated reduced activity in PIGK R60A]. The results (new **Fig. 5f, Fig. S10**), as quoted below, are largely consistent with the auto-inhibition model.

“To further test the auto-inhibition model, we designed three mutations to destabilize the inactive state: PIGK H235A, H244A, and R248A. Additionally, we introduced PIGK H235F to not only destabilize the inactive state by eliminating the H-bond with GPAA1 Ser53 but also stabilize the active state by forming hydrophobic interactions with PIGK Phe251/Tyr252 (**Fig. 5e**). We anticipate that these mutants would increase the apparent activity and show higher surface staining of GPI-AP markers compared to the wild-type PIGK. However, FACS results with the endogenous marker CD59 did not show differences among the PIGK mutants and the wild-type (**Supplementary Fig. 10a**). This result was challenging to interpret, as the surface display of CD59 is influenced by enzymes and transporters in the entire GPI-AP biogenesis pathway. Moreover, it is unclear whether GPI-T is the rate-limiting enzyme in this context. Nevertheless, one straightforward explanation for the lack of differences could be that the limited availability of the endogenous CD59 proprotein becomes a limiting factor for the cell-based FACS assay, thus failing to report the full potential of gain-of-function mutants.

To overcome the substrate availability issue, we modified the FACS assay by introducing an overexpressed GPI-AP reporter. A chimera TGP- and CD55-based GPI-AP (dubbed CD55*, **Supplementary Fig. 2**) was constructed similarly to the pULBP2^{chimera} used in GPI-T^{prod}. To differentiate GPI-AP-expressing cells (GFP fluorescence) from PIGK-expressing cells, we added a mCherry tag to PIGK. GPI-T activity was assessed in PIGK-KO cells by fluorescence gating for CD55-TGP

expression, PIGK expression, and the surface display of CD55* (via Flag-tag). Although the apparent GPI-T activity for PIGK H235A, H244A, and R248A were similar to that of the wild-type, cells transfected with PIGK H235F exhibited higher fluorescence intensity (**Fig. 5f, Supplementary Fig. 10b**). This result is consistent with the abovementioned double-action design for PIGK H235F. The extent to which other mutants also promote GPI-T activity remains to be investigated using more quantitative and, preferably, test-tube biochemical assays. ”

Figure 5 | Activation of the GPI-T complex requires an energetically unfavorable conformational change of an autoinhibitory loop.

e Energetically unfavorable rearrangement of four sidechains (R60/H244/D247/R248) associated with the conformational change of the 231-Loop. The inactive state (cyan) is stabilized by H-bonds, ionic locks, and cation- π interactions, which are broken upon activation (magenta) and replaced with unfavorable interactions such as electrorepulsion. The N-to-C direction of $\alpha 1$ is indicated by a black arrow, and the positive end of the helix dipole is labeled as δ^+ . **f** Apparent activity of GPI-T mutants containing substitutions of PIGK residues implicated in the auto-inhibition mechanism. The surface expression of a chimera GPI-AP (CD55*, **Supplementary Fig. 2**) in PIGK-KO cells transfected with the wild-type PIGK (cyan), PIGK mutants (red), or an irrelevant membrane protein (grey) was assessed by FACS. Typical results of three independent experiments are shown.

Supplementary Figure 10 | Apparent activity of mutant GPI-T containing substitutions of PIGK residues implicated in the auto-inhibition mechanism. a

The surface expression of the endogenous GPI-AP marker (CD59) in PIGK-KO cells transfected with the wild-type PIGK (cyan), PIGK mutants (red), and an irrelevant membrane protein (grey) was assessed by fluorescence-activated cell sorting (FACS).

b The same as in **a** except that the marker was an overexpressed chimera GPI-AP (CD55*, **Supplementary Fig. 2**).

Reviewer #2

GPI anchoring is a highly conserved mechanism in eukaryotic cells, and this post translational modification is critical in various biological processes. The addition of the GPI to the proprotein is carried out by the GPI-T complex of proteins, which consists of five subunits. The GPI-T complex exhibits broad specificity recognizing a broad pattern of C-terminal peptides. Using protein and cell engineering strategies the authors successfully purified and solved the structure of the GPI-T complex in its substrate bound form as well as in its product bound form. To solve the structure of the substrate bound form the authors mutated C206S in PIGK, so that the enzyme complex still binds proproteins, but the protein is catalytically dead. To capture the product bound state the authors genetically engineered a cell line to disrupt the gene encoding GPI-AP deacylase which is downstream of the GPI-T complex.

I overall find the protein and genetic engineering to be cleverly done, and the structural work good and I would recommend publishing the paper.

I have some suggestions for changes to the text and a couple of sections that needs more details or explanation.

We thank the reviewer for the specific comments to improve our manuscript and support review of our work.

Major points:

I think the surface staining experiments are not well described in the text. For example, line 113-115 is assuming the reader already know the assay and if you don't, you need to search in the methods to understand what the assay is, the authors don't even mention that this is fluorescent activated cell sorting in the text.

We thank the reviewer for the suggestion to increase clarity. The FACS assay has now been introduced in the first section of the Results.

“...Additionally, we introduced a hemagglutinin (HA)-tag for detecting ULBP2

surface expression by fluorescence-activated cell sorting (FACS) (**Supplementary Fig. 2**). While most GPI-APs depend on GPI-anchoring for surface display, and FACS signals are lost with an incompetent ω -site...”

Also, I would like an introduction to CD59 and prion labeling, these are mentioned and used, but never introduced.

An introductory sentence was added for the two GPI-AP markers. By the way, we have changed “prion” to “prion protein (PrP)” throughout the text because prion is typically used to describe the misfolded and infectious form of PrP.

“To test functional relevance of the structural observation, we created mutations and tested their apparent activity using the aforementioned FACS assay with two endogenous GPI-AP markers: CD59, a membrane complement regulator that inhibits the formation of the membrane attack complex, and prion protein (PrP), a glycoprotein that causes prion diseases when misfolded.”

It’s unclear to me why the GFP fusion was used?

A thermostable GFP (TGP) chimera was used because the expression level of the chimera GPI-AP is generally higher than the original ones based on our experiences.

The paragraph was expanded to make this point clearer.

“Initial attempts to purify the co-expressed GPI-T and ULBP2 using the same affinity purification strategies as GPI-T^{sub} yielded an insufficient amount (60 μ g per liter of culture) of the product complex, mainly due to a low yield (4%) during the second affinity chromatography step. The low yield suggested weaker binding of the products to the enzyme compared with substrates. We reasoned that an artificial proprotein with a stable core, such as the thermostable green fluorescence protein (TGP)³⁶, may express at a higher level than ULBP2 and thus promote the formation of the enzyme-product complex. Moreover, a fluorescent GPI-AP would allow for convenient

assessment of its stoichiometry to the TGP-tagged GPI-T subunits through in-gel fluorescence³⁶. Therefore, we constructed a fluorescent proprotein called proULBP2* by grafting the N-/C-terminal signal peptide and the ω-9 region of proULBP2 onto TGP³⁶ (**Supplementary Fig. 2**). The PI-PLC sensitivity assay confirmed successful GPI-anchoring of the chimera protein (**Supplementary Fig. 3d**). Subsequently, we co-expressed and co-purified proULBP2* with the wild-type GPI-T. As expected, the yield for the enzyme-ULBP2* complex (dubbed GPI-T^{prod}) increased by 3.1-fold compared with the enzyme-ULBP2 complex. Gel filtration fractions (**Supplementary Fig. 3e**) showing approximately equimolar amounts of GPI-T subunits and ULBP2* on in-gel fluorescence (**Supplementary Fig. 3f**) were used for structural analysis.”

And I would like an illustration of what this construct looks like, I have a hard time understanding the sentence describing the complex in line 147-149.

We thank the reviewer for the suggestion on the illustration and we are sorry for the confusion caused.

An illustration has been included in **Supplementary Fig. 2**.

And **Supplementary Fig. 2** is now cited wherever appropriate to increase readability.

Supplementary Figure 2 | Constructs and processing of recombinant GPI-APs in this study. **a** Schematic of elements in the recombinant GPI-APs. The name and their purpose are indicated above each construct, while figures citing the constructs are indicated in blue texts. An internal ribosome entry site (IRES) in (*i*) and (*iv*) is used to co-express the GPI-AP and tagBFP, which serves as a gating marker during fluorescence-activated cell sorting (FACS) for the successful expression of GPI-APs. BFP, blue fluorescence protein; GS, glycine-serine linker; HA, hemagglutinin tag; TGP, thermostable green fluorescence protein. **b** Schematic of the GPI-APs in **a** and their speculated cellular fate. Relevant residues numbering are shown. The glycan part of GPI is indicated by magenta pentagons. The acyl chains of GPI are colored

magenta, green (inositol acyl), or black (remodeled *sn-2*). Figures citing the GPI-APs are indicated by blue text. A “+/-” sign in **(iii)** indicates compromised vesicle transport of the triacylated GPI-AP due to the absence of PGAP1. ER, endoplasmic reticulum.

I would like a little more detail in the description of the purification of the complex, you can find it in the methods, but I think a couple of sentences in the main text would be useful (line 135-138).

We thank the reviewer for the suggestion. We have expanded the purification details as follows:

“To ensure the proprotein’s integrity, a dead GPI-T mutant³² (PIGK C206S) which still binds proproteins²⁷ was co-expressed with His-tagged proULBP2 (**Supplementary Fig. 2**) in PIGK knockout (KO) HEK293 cells³³. The inactive enzyme (GPI-T^{C206S}) and proULBP2 were then co-purified by tandem affinity chromatography using a Strep-tag on the PIGU subunit of the GPI-T complex and a His-tag on proULBP2. The relative yield of the second affinity chromatography over the first was approximately 40%, suggesting a relatively tight substrate-bound complex. Consistently, gel filtration (**Supplementary Fig. 3b**) and SDS-PAGE (**Supplementary Fig. 3c**) showed co-elution of the Michaelis complex (GPI-T^{sub}) as a symmetric and monodisperse peak. This purification scheme yielded 0.5 mg of GPI-T^{sub} per liter of culture.

To capture an enzyme-product complex, we hypothesized that impairing downstream GPI-AP maturation (**Supplementary Fig. 1a**) could enhance the transamidase’s ability to efficiently hold its product(s), thereby enabling the co-purification of the enzyme-product(s) complex. To test this hypothesis, we genetically preserved the inositol acyl chain (**Supplementary Fig. 1b**), which is likely required for efficient binding with GPI-T, by disrupting *pgap1*, a gene encoding a GPI-AP deacylase³⁴. This disruption is known to affect downstream vesicle transport and GPI-AP remodeling³⁵, which, in turn, detains ULBP2 in the ER membrane and further facilitates enzyme-product binding.

Initial attempts to purify the co-expressed GPI-T and ULBP2 using the same affinity purification strategies as GPI-T^{sub} yielded an insufficient amount (60 µg per liter of culture) of the product complex, mainly due to a low yield (4%) during the second affinity chromatography step. The low yield suggested weaker binding of the products to the enzyme compared with substrates. We reasoned that an artificial proprotein with a stable core, such as the thermostable green fluorescence protein (TGP)³⁶, may express at a higher level than ULBP2 and thus promote the formation of the enzyme-product complex. Moreover, a fluorescent GPI-AP would allow for convenient assessment of its stoichiometry to the TGP-tagged GPI-T subunits through in-gel fluorescence³⁶. Therefore, we constructed a fluorescent proprotein called proULBP2* by grafting the N-/C-terminal signal peptide and the ω-9 region of proULBP2 onto TGP³⁶ (**Supplementary Fig. 2**). The PI-PLC sensitivity assay confirmed successful GPI-anchoring of the chimera protein (**Supplementary Fig. 3d**). Subsequently, we co-expressed and co-purified proULBP2* with the wild-type GPI-T. As expected, the yield for the enzyme-ULBP2* complex (dubbed GPI-T^{prod}) increased by 3.1-fold compared with the enzyme-ULBP2 complex. Gel filtration fractions (**Supplementary Fig. 3e**) showing approximately equimolar amounts of GPI-T subunits and ULBP2* on in-gel fluorescence (**Supplementary Fig. 3f**) were used for structural analysis.”

Minor points:

I think it would make sense to call it the GPI-T complex throughout the text and not just GPI-T.

We have changed “GPI-T” to “the GPI-T complex” in several places but not all because

- 1) Depending on the context, the use of the latter may cause ambiguity, as “complex” was sometimes used to describe the substrates-enzyme or products-enzyme complex;
- 2) While the use of “the GPI-T complex” is more accurate, we think the use of “GPI-T” is also acceptable, as in the case of γ -secretase, NADH dehydrogenase etc.

Line 60-64 is very confusing. I believe the “is” on line 60 should be a “was”,

By “The GPI-T complex is a promiscuous enzyme” we meant that it is a well-known fact. Therefore we used “is” instead of “was”.

and maybe there should be a “is the” between EtNP2 and preferred instead of the “a” on line 61.

Point taken with thanks.

But the section could use a re-write.

We have rewritten the section as follows:

“The GPI-T complex is a promiscuous enzyme. First, GPI-T exhibits broad proprotein specificity. For example, the human GPI-T complex can process over 150 proproteins ranging from <20 to >2,000 residues. GPI-T recognizes proproteins through a remarkably vague pattern in the C-terminal signal peptide (CSP) region rather than consensus sequences. The pattern consists of an ω -site where GPI is later added, followed by an $\omega+1$ site that can be any residue except proline, a small $\omega+2$ residue, a generally hydrophilic spacer with 8-12 residues, and a stretch of 15-20 hydrophobic residues for membrane association. An unstructured linker of approximately 10 polar residues ($\omega-10$ to $\omega-1$) preceding the CSP is also found in GPI-APs (**Fig. 1a**)¹⁻⁴. The ω -site typically contains residues with small side chains, such as glycine, alanine, serine, asparagine, and aspartate¹⁻⁴, but is recently¹³ expanded to two slightly larger residues (leucine, methionine) and at a C β -branched amino acid (threonine), albeit at a lower frequency (**Fig. 1a**). Second, GPI-T exhibits promiscuity for the GPI substrate. For example, EtNP3 on GPI is long thought to be the sole physiological bridge for GPI attachment, but a recent study¹⁴ shows EtNP2 is the preferred choice for some GPI-APs. What is more, GPI-T exhibits activities with non-GPI amines including hydroxylamine¹⁵. Finally, GPI-T can digest proproteins without GPI attachment¹⁶. Given GPI-T’s broad proprotein specificity, its promiscuity raises an important question as how its activity is safeguarded to prevent unintentional

cleavage.”

For line 106-109 I find the sentence very hard to read and understand, I would consider a re-write.

The section has been re-written as follows:

“The human GPI-T complex can process over 150 different proprotein substrates. To capture GPI-T with a proprotein, selecting a proprotein with a relatively higher affinity for the enzyme was desirable. Considering this, we identified the UL16 binding protein 2 (ULBP2) as a suitable candidate because it was previously co-purified with GPI-T^{26,27}. ULBP2 is a major histocompatibility complex class I-related GPI-AP that activates natural killer cells through binding with the NKG2D receptor²⁸. In a previous study²⁹, Ser216 of ULBP2 was assigned as the ω -site, with Ser217 as a possible alternative. To determine the exact ω -site, we conducted further investigation through mutagenesis.”

Also on line 109-110, Ser216 and Ser217 of what protein?

They are for ULBP2. The information has been included in the revised manuscript.

On line 248 the reference to Fig. 1a does not make sense.

Fig. 1a has been replaced with a reference in the revised manuscript.

In figure 3c I think there is a typo it says Q35P in the left plot.

Corrected with thanks.

First section of the discussion is hard to understand, should it be “committed” in line 503?

The term “committed” was meant to highlight the fact that a protein only becomes a GPI-AP after the reaction of GPI-T. However, one might argue that the enzyme responsible for the addition of the acetylglucosamine to phosphatidylinositol catalyzes the committed step for GPI synthesis. Therefore, we have removed this word in the revised manuscript.

The section has been revised as quoted below.

“The human GPI-T complex catalyzes the essential GPI attachment step for the biosynthesis of over 150 GPI-APs. Despite its importance, the mechanism by which the CSP region, lacking an apparent sequence consensus, controls GPI-T activity and how it maintains a balance between broad substrate specificity and fidelity has been a longstanding mystery. In this study, we present the structures of the Michaelis complex of GPI-T, as well as the enzyme-products complex. These structures reveal that the previously reported proprotein-free GPI-T structures^{24,25} exist in an inactive state, while the current structures represent the active state. Through mutagenesis and structural analyses, we demonstrate that seemingly weak features of the CSP region collectively form a strong selective filter for the activation of GPI-T, elucidating how this region controls substrate suitability despite the low sequence consensus. Furthermore, the architecture of the proprotein-binding site rationalizes GPI-T’s broad substrate specificity and its moderate selectivity of ω -site residues. The structures also suggest caspase-like catalytic mechanisms for substrate activation and catalysis.”

On line 504 an “are” is missing between “here” and “the”

The whole paragraph has been rewritten as quoted above.

On line 509 to 510 it should read: ... and help to boost accuracy of existing algorithms.

Done.

Reviewer #3

Glycosylphosphatidylinositol transamidase (GPI-T), a complex of five transmembrane proteins, catalyzes posttranslational modification of many eukaryotic cell surface proteins by GPI-anchors. The enzyme recognizes a GPI attachment signal peptide located at the C-terminus of proproteins (CSP) and replaces it with a preassembled GPI that acts as membrane anchors of the modified protein. More than 150 human proteins are GPI-anchored, however, sequences of their CSPs are different and have only several common characteristics. A remarkable feature of GPI-T is its ability to recognize all those only weakly conserved sequences and faithfully attaches GPI. This research group previously reported 3D structure of human GPI-T with associated GPI but lacking proprotein, termed GPI-T_{apo}. To clarify structural basis of transamidation-mediated GPI attachment, authors took a clever approach and were able to purify two forms of GPI-T, GPI-T_{sub} bearing two substrates (GPI and a proprotein), and GPI-T^{prod} bearing two products (a GPI-anchored protein and a CSP). Comparing 3D structures of those three GPI-Ts determined by cryoEM, authors report several important points relevant to understanding the GPI attachment mechanism: (1) CSP makes functionally important association with sole TMD of PIGT.; (2) GPI associates with GPI-T through interactions of lipid moiety with PIGU, interactions of three ethanolamine phosphates with GPAA1, and interactions of juxtamembrane region with PIGT and PIGU.; (3) Presence of a deep pocket accommodating only a small amino acid at the omega site (to which GPI is attached) and shallow grooves connecting to both sides of the deep pocket accounting for accommodation of CSP of various amino acid sequences.; (4) A fishing rod-like mechanism of GPI attachment to the C-terminus of the omega site; (5) Demonstration of inactive state of GPI-T, in which the catalytic region is covered by an auto-inhibitory loop of PIGK.; (5) Activation of GPI-T by flipping out of the auto-inhibitory loop and its rearrangement to generate environment of the catalytic region.; and (6) Possible mechanism that prevents unintentional cleavage.

Based on these findings, authors proposed a model of GPI attachment reactions. The model is supported by structural data and explains unique features of GPI-T actions. These findings greatly advance understanding of mechanisms of GPI attachment.

The paper is clearly written and is easy to read. I have only minor points and corrections.

We thank the reviewer for the careful evaluation and the valuable suggestions to improve our manuscript.

Minor points.

1. Fig1a, c and d. GPI is illustrated as having four mannoses in Fig 1a whereas GPI in GPI-T^{sub} and GPI-T^{prod} had three mannoses. It would be better if illustration and actual structures are consistent.

The Mannose 4 (Man4) is optional for the human GPI-T and the majority of the GPI-APs in human does not have Man4. We have modified **Fig. 1a** with a sign of (+/-) beside Man4 to reflect this point. The information has also been added to the figure legends:

“A “+/-” sign indicates the optional Man4 modification for human GPIs.”

2. Fig. 3 and Fig. 4. It would be informative for readers if authors comment on whether fourth mannose can be accommodated in GPI-T^{sub} without affecting the local structure and whether a fishing-rod-like movement can occur with the fourth mannose.

We thank the reviewer for the insightful comment. We have added the following texts in the revised manuscript.

“It is worth noting that, despite the well-defined density for the glycans and EtNPs, there was no apparent evidence for a fourth mannose (Man4). This observation is consistent with previous findings⁴⁴ that the mammal GPIs rarely contain Man4. However, it is important to highlight that GPI-T can still accommodate Man4, as the Man3 2-hydroxyl where Man4 may be added points to the bulk solvents (**Fig. 3a**).”

Figure 3 | GPI binds GPI-T with a rich network of interactions. **a** Interactions between GPI (stick) and GPI-T^{C206S} (cylinder-cartoon with interacting residues in stick representations). The cryo-EM density of GPI is represented by a transparent grey surface. H-bonds/metal coordination with distances in Å are indicated by dash lines. GPI is colored alternatingly for better visualization. The catalytic dyad residue His164 (blue sphere) and proULBP2 (green cartoon) are shown for orientation purposes. An asterisk on Man3 indicates the 2-hydroxyl where a fourth mannose may be added. GlcN, glucosamine; Ino, inositol; TMH, transmembrane helix.

“The 2-hydroxyl of Man3, where Man4 is infrequently added in mammal GPIs⁴⁴, undergoes an approximately 180 °flip during the “fishing-rod” movement (although we acknowledge that the accuracy of Man3/EtNP3 is affected by the less-defined density in GPI-T^{prod}). This flipping motion brings Man4 from an open space to a cleft

between PIGK and GPAA1 (Fig. 4d), potentially causing clashes. These clashes could be part of the mechanism responsible for the infrequent occurrence of Man4 in mammal GPI-APs. On the other hand, it is also plausible that steric hindrance may not be an issue due to the flexibility of the mannoses/EtNPs and the spacious local environment.”

Figure 4 | A fishing rod-like mechanism for GPI attachment.

d A fishing rod-like movement of GPI from the position in GPI-T^{sub} to the superimposed position in GPI-T^{prod}. The subunits, ULBP2* (alternating coloring), CSP (dark green) are taken from GPI-T^{prod}. ProULBP2 (bright green) and GPI (grey) are superposed from GPI-T^{sub}. An asterisk on Man3 indicates the 2-hydroxyl where a fourth mannose may be added. The inset illustrates the movement of Man3 from the state in GPI-T^{sub} to that in GPI-T^{prod}, viewed at a different angle than the main figure.

3. p3, line 44. GPI biosynthesis insufficiency has not been reported in cancers. Amplification and upregulation of genes of GPI-T components were reported in

cancers. The sentence needs to be modified for accuracy

Done. We thank the reviewer for catching this error.

Other corrections.

1. p3, line 45. Hepatic not haptic?

Done.

2. p3, line 52. Glucosamine not glycosylamine.

Done.

3. p4, line 55. Later rather than latter?

Done.

4. p20, line 442. PIGS not PGIS.

Done. We thank the reviewer for the careful reading.

REVIEWERS' COMMENTS

Reviewer #1 (Remarks to the Author):

The authors have satisfactorily addressed all of the points that were raised in my initial review. I support its publication at Nature Communications.